# Fas signaling-mediated $T_H9$ cell differentiation favors bowel inflammation and antitumor functions

Yingying Shen[1,2,3,14], Zhengbo Song[4,14], Xinliang Lu[1,14], Zeyu Ma[1], Chaojie Lu[1], Bei Zhang[1], Yinghu Chen[5], Meng Duan[6], Lionel Apetoh [7,8], Xu Li[9], Jufeng Guo[10], Ying Miao[11], Gensheng Zhang[12], Diya Yang[13], Zhijian Cai[3] & Jianli Wang[1,2]

Fas induces apoptosis in activated T cell to maintain immune homeostasis, but the effects of non-apoptotic Fas signaling on T cells remain unclear. Here we show that Fas promotes $T_H9$ cell differentiation by activating NF-κB via $Ca^{2+}$-dependent PKC-β activation. In addition, PKC-β also phosphorylates p38 to inactivate NFAT1 and reduce NFAT1-NF-κB synergy to promote the Fas-induced $T_H9$ transcription program. Fas ligation exacerbates inflammatory bowel disease by increasing $T_H9$ cell differentiation, and promotes antitumor activity in p38 inhibitor-treated $T_H9$ cells. Furthermore, low-dose p38 inhibitor suppresses tumor growth without inducing systemic adverse effects. In patients with tumor, relatively high $T_H9$ cell numbers are associated with good prognosis. Our study thus implicates Fas in $CD4^+$ T cells as a target for inflammatory bowel disease therapy. Furthermore, simultaneous Fas ligation and low-dose p38 inhibition may be an effective approach for $T_H9$ cell induction and cancer therapy.

[1] Institute of Immunology and Bone Marrow Transplantation Center of the First Affiliated Hospital, Zhejiang University School of Medicine, 310058 Hangzhou, China. [2] Institute of Hematology, Zhejiang University and Zhejiang Engineering Laboratory for Stem Cell and Immunotherapy, 310003 Hangzhou, China. [3] Institute of Immunology and Department of Orthopaedics of the Second Affiliated Hospital, Zhejiang University School of Medicine, 310058 Hangzhou, China. [4] Department of Medical Oncology, Zhejiang Cancer Hospital, 310022 Hangzhou, China. [5] Division of Infection Disease, Zhejiang Key Laboratory for Neonatal Diseases, Children's Hospital, Zhejiang University School of Medicine, 310006 Hangzhou, China. [6] Chronic Disease Research Institute, School of Public Health, School of Medicine, Zhejiang University, 310058 Hangzhou, China. [7] INSERM, U866 Dijon, France. [8] Faculté de Médecine, Université de Bourgogne, Dijon 21000, France. [9] School of Life Science, Westlake University, 310024 Hangzhou, China. [10] Department of Breast Surgery, Affiliated Hangzhou First People's Hospital, Zhejiang University School of Medicine, 310006 Hangzhou, China. [11] Clinical Trial Center, Qingdao Municipal Hospital, 266011 Qingdao, China. [12] Department of Critical Care Medicine, Second Affiliated Hospital, Zhejiang University School of Medicine, 310009 Hangzhou, China. [13] Xinyuan Institute of Medicine and Biotechnology, School of Life Sciences, Zhejiang Sci-Tech University, 310018 Hangzhou, China. [14]These authors contributed equally: Yingying Shen, Zhengbo Song, Xinliang Lu [15]These authors jointly supervised this work: Zhijian Cai, Jianli Wang. Correspondence and requests for materials should be addressed to Z.C. (email: caizj@zju.edu.cn) or to J.W. (email: jlwang@zju.edu.cn)

Receiving specific patterns of cytokine signaling during activation, CD4$^+$ T cells will differentiate into different effector T cell subsets. Interleukin-12 (IL-12) promotes T helper type 1 (T$_H$1) cell differentiation. IL-4 promotes T$_H$2 cell differentiation. Transforming growth factor-β1 (TGF-β1) and IL-6 stimulate T$_H$17 cell differentiation, while TGF-β1 alone drives T regulatory cell (Treg) differentiation[1–3]. IL-9-producing T$_H$9 cells were first identified by reprogramming T$_H$2 cells with TGF-β1 or Foxp3$^+$ Tregs with IL-4[4,4]. Currently, it is generally accepted that IL-4 and TGF-β1 can induce T$_H$9 cell differentiation[6]. T$_H$9 cells are reported to exacerbate autoimmune and allergic diseases[7–10]. They also exhibit robust antitumor activity superior to that of T$_H$1 and T$_H$17 cells[11,12]. A recent publication demonstrated that one transfer of T$_H$9 cells is sufficient to eradicate advanced tumors[13], indicating the promise of T$_H$9 cells in adoptive cancer therapy. Therefore, it is necessary to further elucidate the underlying mechanism of T$_H$9 cell differentiation.

Accumulating evidence has unveiled the regulation of T$_H$9 cell differentiation at the molecular level. CD4$^+$ T cells with interferon-regulatory factor 4 (IRF4), GATA-3, or STAT6 deficiency fail to develop into T$_H$9 cells[5,8,14], indicating the essential roles of these transcription factors (TFs) in T$_H$9 cell differentiation. PU.1 is also required for the development of T$_H$9 cells[9]. The cooperative signaling among the TGF-β1-activated kinase TAK1, STAT5, Notch, Smad, and RBP-Jκ participates in T$_H$9 cell differentiation[15,16]. OX40 and glucocorticoid-induced tumor necrosis factor receptor (TNF-R)-related protein, two members of the TNF-R superfamily, induce T$_H$9 cells by activating the nuclear factor-κB (NF-κB) pathway[17,18]. Moreover, DR3, another member of the TNF-R family, enhances T$_H$9 cell differentiation through a STAT5-dependent mechanism[19]. Whether other TNF-R family molecules regulate T$_H$9 cell differentiation has yet to be explored.

Fas, a member of the TNF-R family, plays a critical role in programmed cell death[20]. Activation-induced cell death (AICD), mediated by the interaction of Fas and Fas ligand (FasL), is important in maintaining T cell homeostasis[21]. Fas is also necessary for T cell proliferation and activation[22,23]. A recent publication demonstrated that Fas promotes T$_H$17 cell differentiation and inhibits T$_H$1 cell development[24]. However, whether non-apoptotic Fas signaling participates in regulating T$_H$9 cell differentiation remains unknown.

Here, we demonstrate that Fas activates protein kinase Cβ (PKCβ) in a Ca$^{2+}$-dependent manner. PKCβ then induces NF-κB activation and, in cooperation with NFAT1, NF-κB-mediated T$_H$9 cell differentiation. By contrast, PKCβ-activated p38 inactivates NFAT1, thereby limiting Fas-mediated T$_H$9 cell differentiation. Fas ligation-induced T$_H$9 cells (FasL-T$_H$9) excerbates inflammatory bowel disease (IBD). In parallel, low-dose p38 inhibitor restores Fas-mediated T$_H$9 cell induction in vitro and in vivo, and greatly suppresses tumor progression via IL-9 without inducing systemic adverse effects. Thus, our study reveals crucial functions of Fas-induced non-apoptotic signaling in T$_H$9 cell induction, and may have important clinical implications in autoimmune disease and cancer therapy.

## Results

**Fas signaling promotes T$_H$9 cell differentiation in vitro**. To determine the role of Fas signaling in T$_H$9 cell differentiation, we differentiated naive CD4$^+$ T cells from wild-type (WT) and *Fas$^{lpr}$* mice (termed WT or *Fas$^{lpr}$* CD4$^+$ T cells, respectively) into T$_H$1, T$_H$2, T$_H$9, and T$_H$17 cells and Tregs in vitro. The *Fas$^{lpr}$* CD4$^+$ T cells generated substantially lower levels of IL-9-producing cells and IL-9 protein than did the WT CD4$^+$ T cells (Fig. 1a, b). Consistent with a previous publication[24], increased T$_H$1 and

decreased T$_H$2 and T$_H$17 cell differentiation could be observed with the *Fas$^{lpr}$* CD4$^+$ T cells (Fig. 1a, b). However, Treg differentiation was significantly inhibited in our polarization systems (Fig. 1a). Changes in signature cytokines or TF messenger RNA (mRNA) levels between the WT and *Fas$^{lpr}$* CD4$^+$ T cells also confirmed these results (Fig. 1c). T$_H$9 cells differentiated from the WT or *Fas$^{lpr}$* CD4$^+$ T cells (termed WT-T$_H$9 and *Fas$^{lpr}$*-T$_H$9, respectively) expressed high levels of the genes encoding the T$_H$9-related TFs PU.1 (*Spi1*) and IRF4 (*Irf4*) (Supplementary Fig. 1a). Nevertheless, the gene levels of the T$_H$1-, T$_H$2-, T$_H$17-, and Treg-related TFs T-bet (*Tbx21*), GATA-3 (*Gata3*), RORγt (*Rorc*), and Foxp3 (*Foxp3*), respectively, and the corresponding signature cytokines IFN-γ (*Ifng*), IL-4 (*Il4*), and IL-17A (*Il17a*) were very low in these cells (Supplementary Fig. 1a). These results demonstrated that Fas signaling enhanced, but did not skew, T$_H$9 cell differentiation. T$_H$9 differentiation in WT CD4$^+$ T cells with Fas knockdown was also blunted (Supplementary Fig. 1b–d), excluding the possibility that the decreased T$_H$9 cell-polarizing ability resulted from developmental defect in the *Fas$^{lpr}$* CD4$^+$ T cells. To further confirm the role of Fas signaling in T$_H$9 cell differentiation, we ligated Fas on WT CD4$^+$ T cells with anti-Fas (Jo2) in vitro and found that Jo2 markedly increased the frequency of IL-9-producing T cells and the IL-9 protein and mRNA levels (Fig. 1d–f). As Fas signaling mediates T cell apoptosis[25,26], we detected the apoptosis of T$_H$9 cells with or without Jo2 stimulation. We found that Jo2 did not affect T$_H$9 cell apoptosis (Supplementary Fig. 1e). Moreover, Jo2 did not alter T$_H$9 cell proliferation (Supplementary Fig. 1f). Fas signaling-induced AICD is the essential mechanism that maintains T cell homeostasis[21]. To assess AICD in FasL-T$_H$9, T$_H$9 cells induced by conventional or Fas ligation methods were restimulated with anti-CD3 and anti-CD28. Compared with the conventionally differentiated T$_H$9 cells (cT$_H$9), the FasL-T$_H$9 showed increased AICD (Supplementary Fig. 1g). However, the FasL-T$_H$9 secreted more IL-9 than the cT$_H$9 (Supplementary Fig. 1h). These results further proved that Fas signaling promotes induction of IL-9-producing T cells.

To test whether autoactivated Fas signaling can increase IL-9-producing T cell numbers, we detected *Fas* and *Fasl* mRNA levels in T$_H$ cell subsets and found that T$_H$9 cells had lower *Fas* but higher *Fasl* gene expression than the other T$_H$ subsets (Supplementary Fig. 1i, j). The ratio of the *Fas* gene level to the *Fasl* gene level was highest in the T$_H$9 cells (Supplementary Fig. 1k), suggesting that autoactivated Fas signaling may play an important role in the induction of IL-9-producing T cells. Then, we differentiated naive CD4$^+$ T cells from WT and *Fasl$^{gld}$* mice into T$_H$9 cells in vitro. FasL deficiency greatly inhibited the induction of IL-9-producing T cells (Fig. 1g–i), and a similar inhibitory effect was also achieved with FasL blocking antibodies (anti-FasL) (Supplementary Fig. 1l). Transfection of a FasL-expressing vector but not an empty vector (EV) rescued the decreased induction of IL-9-producing T cells within the CD4$^+$ T cells from *Fasl$^{gld}$* mice (Supplementary Fig. 1m). These findings indicated that autoactivated Fas signaling reinforces the induction of IL-9-producing T cells in vitro.

**Fas signaling activates genes related to T$_H$9 cell functions**. To better understand the effects of Fas signaling on the T$_H$9 cell program, we performed RNA-sequencing (RNA-seq) analysis of WT-T$_H$9 and *Fas$^{lpr}$*-T$_H$9. There were 204 differentially expressed genes (DEGs) between the WT-T$_H$9 and *Fas$^{lpr}$*-T$_H$9 (Supplementary Data 1). Among these DEGs, 84 genes had upregulated expression, and 120 genes had downregulated expression in the *Fas$^{lpr}$*-T$_H$9 (Fig. 2a). *Il9* was one of the most highly downregulated genes in the *Fas$^{lpr}$*-T$_H$9 (Fig. 2a). *Gzm* genes have been

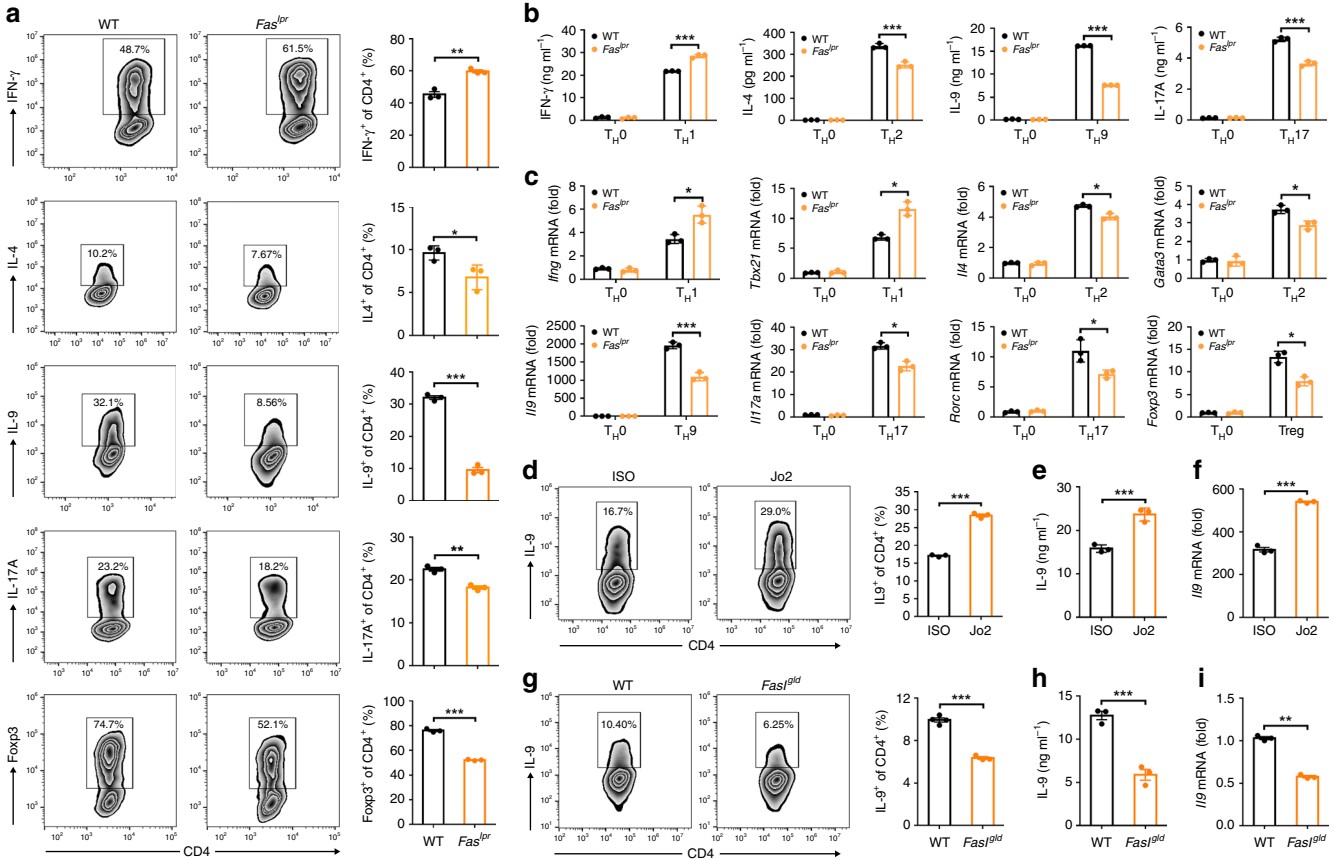

**Fig. 1** Fas signaling promotes T helper type 9 ($T_H9$) cell differentiation in vitro. **a–c** Naïve CD4+CD62LhiCD44lo T cells were sorted from wild-ype (WT) and $Fas^{lpr}$ mice and differentiated into $T_H0$, $T_H1$, $T_H2$, $T_H9$, and $T_H17$ cells and T regulatory cells (Tregs) in the presence of plate-bound anti-CD3 and anti-CD28 for 3–4 days. Flow cytometric analysis of the frequencies of IFN-γ+, IL-4+ (interleukin-4+), IL-9+, IL-17A+, and Foxp3+ cells among CD4+ T cells (left) and the corresponding statistical analysis (right) on day 4 (**a**); enzyme-linked immunosorbent assay (ELISA) measurements of the IFN-γ, IL-4, IL-9 and IL-17A levels in supernatants from the $T_H0$, $T_H1$, $T_H2$, $T_H9$, and $T_H17$ cells on day 3 (**b**); and real-time PCR analysis of the expression of the indicated genes in the $T_H0$, $T_H1$, $T_H2$, $T_H9$, and $T_H17$ cells and Tregs on day 3 (**c**). **d–f** Naive CD4+CD62LhiCD44lo T cells were sorted from WT mice and differentiated into $T_H9$ cells with 10 μg ml$^{-1}$ isotype control antibodies (ISO) or anti-Fas antibodies (Jo2) for 3–4 days. Flow cytometric analysis of the frequency of IL-9+ cells among the CD4+ T cells (left) and the corresponding statistical analysis (right) (**d**), ELISA measurement of the IL-9 cytokine levels in the supernatants of the $T_H9$ cells (**e**), and real-time PCR analysis of $Il9$ gene expression in the $T_H9$ cells (**f**). **g–i** Naive CD4+CD62LhiCD44lo T cells were sorted from WT and $Fas^{gld}$ mice and differentiated into $T_H9$ cells for 3–4 days. Flow cytometric analysis of the frequency of IL-9+ cells among the CD4+ T cells (left) and the corresponding statistical analysis (right) (**g**), ELISA measurement of the IL-9 cytokine levels in the supernatants of the $T_H9$ cells (**h**), and real-time PCR analysis of $Il9$ gene expression in the $T_H9$ cells (**i**). *$P < 0.05$, **$P < 0.01$, and ***$P < 0.001$ (unpaired Student's $t$ test). Representative results from three independent experiments are shown (mean and s.d.) ($n = 3$)

reported to have increased expression in $T_H9$ cells and contribute to the antitumor activity of $T_H9$ cells[13]. We found that $Gzma$, $Gzmc$, $Gzmd$, $Gzme$, and $Gzmg$ were downregulated, but the master regulator of granzyme $Eomes$[27] and $Gzmk$ were upregulated in the $Fas^{lpr}$-$T_H9$ (Fig. 2a), which were confirmed by real-time PCR (Fig. 2b), suggesting $Eomes$-independent regulation of granzyme in $T_H9$ cells. A heat map obtained by unsupervised hierarchical clustering showed that the WT-$T_H9$ and $Fas^{lpr}$-$T_H9$ converged in the same cluster and showed overlapping gene expression (Fig. 2c). Moreover, $T_H1$ cell-, $T_H2$ cell-, $T_H17$ cell-, and Treg-related genes showed no obvious differences between the WT-$T_H9$ and $Fas^{lpr}$-$T_H9$ (Fig. 2d). These results indicated that Fas deficiency does not globally affect $T_H9$ cell differentiation. $T_H9$ cells play a key role in the progression of autoimmune diseases[28]. DEG results from an analysis of the most highly enriched pathways revealed that genes related to autoimmune thyroid disease, type I diabetes mellitus, and systemic lupus erythematosus were downregulated in the $Fas^{lpr}$-$T_H9$ (Fig. 2e). Therefore, RNA-seq analysis demonstrated that Fas signaling

contributes to the gene activation, and is responsible for $T_H9$ cell-specific functions.

**Fas signaling induces $T_H9$ cells by activating NF-κB.** To examine how Fas signaling induces $T_H9$ cells, we first investigated the differentiation of FasL-$T_H9$ in the presence of the pancaspase inhibitor z-VAD-fmk[29] and found that z-VAD-fmk did not alter the Jo2-mediated increase in IL-9-producing T cells (Supplementary Fig. 2a), indicating caspase-independent induction of $T_H9$ cells. Since Fas defects inhibit $T_H9$ cell and Treg differentiation, and IL-2 is critical for both cell differentiation[30,31], we next tested the role of IL-2 in Fas-mediated $T_H9$ cell differentiation. We found that $Il2$ mRNA levels were not different between WT-$T_H9$ and $Fas^{lpr}$-$T_H9$ (Supplementary Fig. 2b). Moreover, neither the addition of exogenous IL-2 nor neutralization of endogenous IL-2 rescued differentiation inferiority of $Fas^{lpr}$-$T_H9$ (Supplementary Fig. 2c, d), indicating IL-2-independent increase of $T_H9$ cell differentiation by Fas. STAT1,

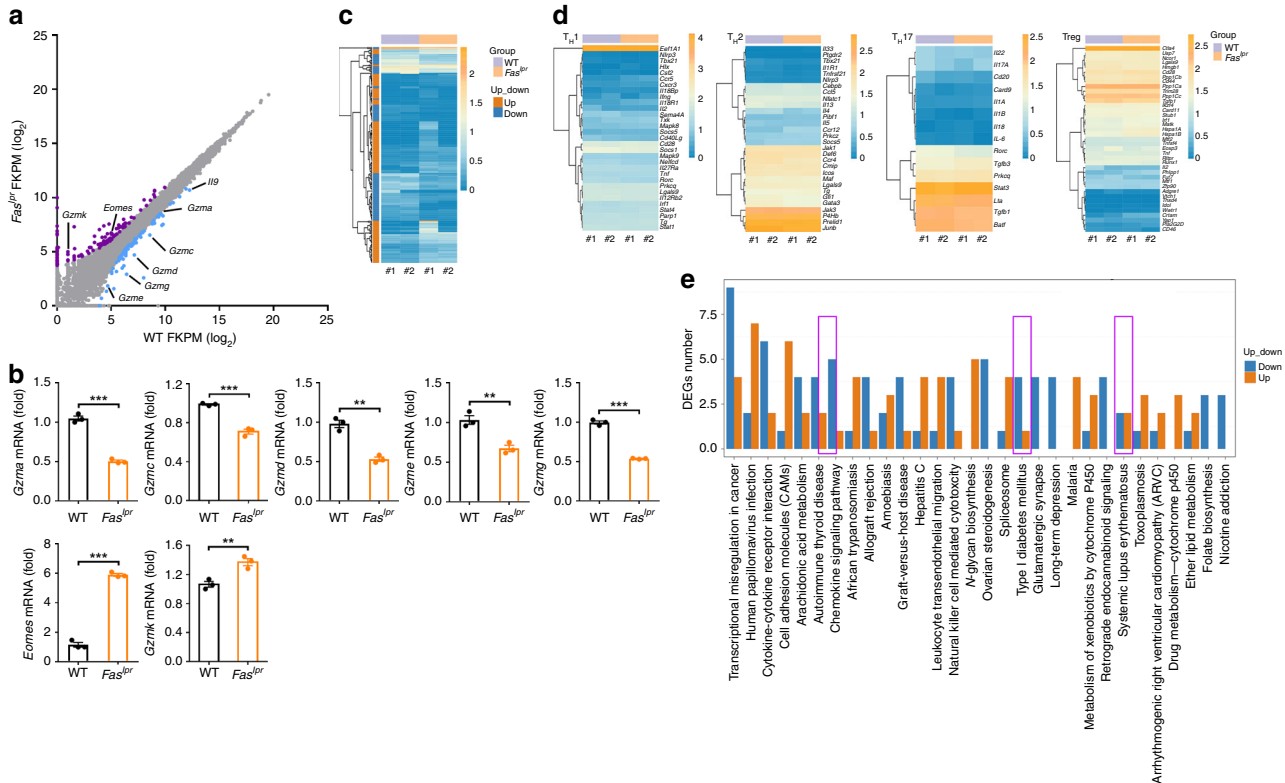

**Fig. 2** Fas signaling activates genes related to T helper type 9 ($T_H9$) cell functions. **a** Scatterplot of RNA-sequencing data showing genes with expression upregulated (purple dots) or downregulated (blue dots) by at least two-fold in $Fas^{lpr}$-$T_H9$ cells relative to wild-type (WT)-$T_H9$ cells and genes with similar expression in $Fas^{lpr}$-$T_H9$ and WT-$T_H9$ cells (gray dots). **b** Real-time PCR analysis of the expression of the indicated genes in $T_H9$ cells differentiated from WT or $Fas^{lpr}$ CD4$^+$ T cells. **c** Heat map of genes expressed in $Fas^{lpr}$-$T_H9$ and WT-$T_H9$ cells. **d** Heat map of $T_H1$ cell-, $T_H2$ cell-, $T_H17$ cell-, and Treg-related genes expressed in $Fas^{lpr}$-$T_H9$ and WT-$T_H9$ cell. **e** Differentially expressed gene (DEG) results for the most highly enriched pathways. Pathways referring to $T_H9$ cell functions are boxed. FKPM, fragments per kilobase of exon per million fragments mapped. $**P < 0.01$ and $***P < 0.001$ (unpaired Student's $t$ test). Representative results from three independent experiments are shown (mean and s.d.) ($n = 3$)

STAT3, STAT5, STAT6, IRF4, PU.1, Gata3, NF-κB, and Akt are all involved in $T_H9$ cell differentiation[6,12,18,32,33]. Mitogen-activated protein kinases also participate in IL-9 expression[34]. We detected the activation and total protein levels of these TFs and kinases in WT and $Fas^{lpr}$ CD4$^+$ T cells cultured under $T_H9$-skewing conditions for 15 min or 24 h and found that the activation of p65 and p38 was obviously inhibited in the $Fas^{lpr}$ CD4$^+$ T cells (Supplementary Fig. 2e, f). In contrast, Fas ligation markedly induced the activation of p65 and p38 (Supplementary Fig. 2e, f). Furthermore, the activation of IKKα, IKKβ, IκBα, p65, and p38 markedly decreased in the $Fas^{lpr}$ CD4$^+$ T cells at different time points (Fig. 3a), indicating Fas signaling-dependent activation of the NF-κB and p38 pathways in differentiating $T_H9$ cells. To test the role of the NF-κB pathways in Fas-mediated $T_H9$ cell differentiation, we treated CD4$^+$ T cells with the IκBα phosphorylation inhibitor BAY 11-7082[35] or the IKK2 inhibitor LY2409881[36] before inducing polarization. Both BAY 11-7082 and LY2409881 completely abolished the differentiation superiority of FasL-$T_H9$ (Fig. 3b), indicating that Fas-induced IL-9-producing T cell generation is NF-κB dependent.

PKC has been reported to participate in the activation of NF-κB[37]. There are three major groups of PKC isoforms. These groups include classic PKCs (PKCα, PKCβ1, PKCβ2, and PKCγ), novel PKCs (PKCδ, PKCε, PKCζ, PKCμ, and PKCθ), and atypical PKCs (PKCζ and PKCι/λ). Fas is known to play a role in PKCβ2 activation[38]. Next, we examined whether PKC is responsible for Fas-induced NF-κB activation. We found that Fas ligation obviously increased the recruitment of PKCβ2 to the plasma

membrane and its colocalization with Fas (Fig. 3c). Increased PKCβ1 recruitment to the plasma membrane and colocalization with Fas were also observed after Fas ligation (Fig. 3c). The selective PKCβ inhibitor enzastaurin[39] and the pan-PKC inhibitor Go 6983 both abrogated the Fas ligation-induced phosphorylation of p65 and increase in IL-9-producing T cells (Fig. 3d, e). These results demonstrated that PKCβ behaves as an activator of NF-κB after Fas ligation.

The activation of classic PKCs is Ca$^{2+}$ dependent[40]. Jo2 stimulation evoked Ca$^{2+}$ flux in WT CD4$^+$ T cells under $T_H9$-skewing conditions (Supplementary Fig. 2g). To define the function of Ca$^{2+}$ in PKCβ activation, we pretreated cells with 2-aminoethoxydiphenyl borate (2-APB) or xestospongin C (XC), which are inhibitors of inositol 1,4,5-trisphosphate-induced Ca$^{2+}$ release[41,42], and found that both prevented PKCβ2 and PKCβ1 recruitment to the plasma membrane and colocalization with Fas in response to Fas ligation (Fig. 3f). Consistently, both inhibitors abolished the activation of p65 and enhancement of IL-9-producing T cell generation (Fig. 3g, h), suggesting that PKCβ activation by Fas ligation is Ca$^{2+}$ dependent.

Fas can trigger Ca$^{2+}$ signaling by activating phospholipase Cγ1 (PLCγ1)[43]. We observed reduced phosphorylated PLCγ1 level in $Fas^{lpr}$ CD4$^+$ T cells under $T_H9$-skewing conditions (Supplementary Fig. 2h). Both U73122, a potent PLC inhibitor[44], and manoalide, an irreversible PLC inhibitor[45], abolished Fas ligation-induced increases in Ca$^{2+}$ flux and IL-9-producing T cells (Supplementary Fig. 2i, j), suggesting that the increased Ca$^{2+}$ response by Fas signaling is PLCγ1-dependent.

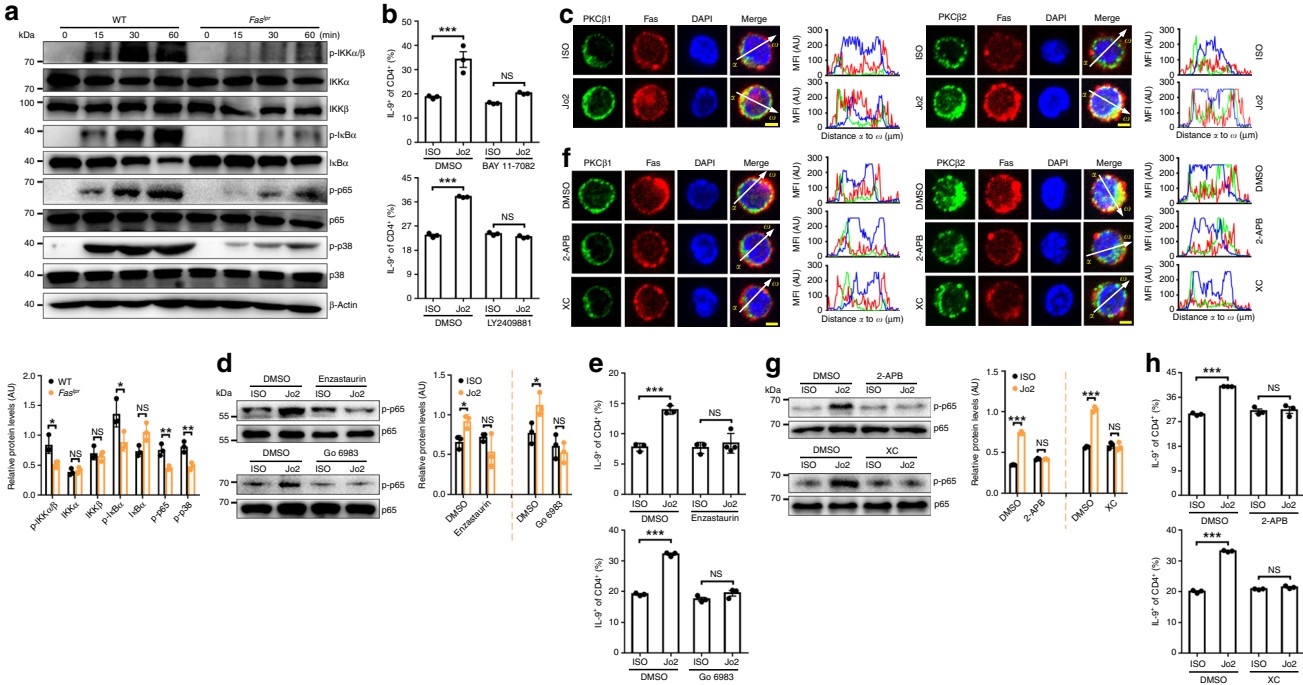

**Fig. 3** Fas signaling induces T helper type 9 ($T_H$9) cell generation by activating nuclear factor-κB (NF-κB). **a** Western blotting analysis of the indicated proteins in wild-type (WT) or $Fas^{lpr}$ CD4$^+$ T cells under $T_H$9-skewing conditions at the indicated time points (top), and statistical analysis of the relative protein levels at the 60 min (bottom). **b** Flow cytometric analysis of the frequency of IL-9$^+$ (interleukin-9$^+$) cells among CD4$^+$ T cells after the stimulation of naive CD4$^+$ T cells with 10 µg ml$^{-1}$ ISO or Jo2 with or without 0.4 µM IκBα inhibitor BAY 11-7082 or 0.5 µM IKK2 inhibitor LY2409881 under $T_H$9-skewing conditions for 4 days. **c** Immunofluorescence staining of protein kinase Cβ (PKCβ1) or PKCβ2 with Fas in naive CD4$^+$ T cells stimulated with 10 µg ml$^{-1}$ ISO or Jo2 for 15 min under $T_H$9-skewing conditions. **d** Western blotting analysis of p-p65 protein levels in naive CD4$^+$ T cells stimulated with 10 µg ml$^{-1}$ ISO or Jo2 for 15 min with or without 0.5 µM PKCβ inhibitor enzastaurin or 0.01 µM pan-PKC inhibitor Go 6983 under $T_H$9-skewing conditions (left), and statistical analysis of the relative protein levels (right). **e** Flow cytometric analysis of the frequency of IL-9$^+$ cells among CD4$^+$ T cells after the stimulation of naive CD4$^+$ T cells with 10 µg ml$^{-1}$ ISO or Jo2 with or without 0.5 µM enzastaurin or 0.01 µM Go 6983 under $T_H$9-skewing conditions for 4 days. **f–h** Immunofluorescence staining of PKCβ1 or PKCβ2 with Fas (**f**), western blotting analysis of p-p65 protein levels (left) and statistical analysis of the relative protein levels (right) (**g**), and flow cytometric analysis of the frequency of IL-9$^+$ cells (**h**) in the naive CD4$^+$ T cells stimulated with 10 µg ml$^{-1}$ Jo2 with or without 10 µM inositol 1,4,5-trisphosphate-induced Ca$^{2+}$ release inhibitor, 2-aminoethoxydiphenyl borate (2-APB) or xestospongin C (XC) for 15 min (**f**, **g**) or 4 days (**h**) under $T_H$9-skewing conditions. Scale bar = 2 µm. AU, arbitrary units; NS, not significant; *$P < 0.05$, **$P < 0.01$, ***$P < 0.001$ (unpaired Student's $t$ test). Representative results from three independent experiments are shown (mean and s.d.) ($n = 3$). Relative protein levels = the indicated protein gray value/β-actin (**a**) or p65 (**d**, **g**) gray value

**Tyr224 and Tyr274 in Fas mediate PLCγ1 activation**. Given that Fas has no enzymatic activity[46], we questioned how PLCγ1 was activated. Zap-70 can interact with Fas and participate in the phosphorylation of PLCγ1 in T cells[47,48]. Fas ligation obviously increased the colocalization of Fas and Zap-70 and phosphorylated Zap-70 level under $T_H$9-skewing conditions (Supplementary Fig. 3a, b). Knocking down Zap-70 expression markedly decreased the Fas ligation-induced phosphorylation of PLCγ1 and generation of IL-9-producing T cell (Supplementary Fig. 3c–f), indicating the Zap-70-dependent phosphorylation of PLCγ1.

There are four Tyr sites in the intracellular domain of Fas (Tyr189, Tyr224, Tyr274, and Tyr284). Tyr284, which is located in the YXXL motif of Fas, is necessary for the binding of Fas with Zap-70[47]. However, a Y284A mutation in Fas did not reduce the colocalization of Fas and Zap-70 (Supplementary Fig. 3g). However, the mutations Y224A and Y274A, but not the mutation Y189A, greatly decreased the colocalization of Fas and Zap-70 (Supplementary Fig. 3g). Consistently, the overexpression of unmutated Fas (Fas-WT) or Fas with the Y189A or Y284A mutation (Fas-Y189A, Fas-Y284A) but not Fas with the Y224A or Y274A mutation (Fas-Y224A, Fas-Y274A) significantly promoted the generation of IL-9-producing T cells from naive $Fas^{lpr}$ CD4$^+$

T cells (Supplementary Fig. 3h). These results demonstrated that both Tyr224 and Tyr274 in Fas are necessary for the Fas ligation-induced generation of IL-9-producing T cells.

**p38 inhibits $T_H$9 cell generation by mediated Fas signaling**. Since Fas ligation obviously induced p38 activation (Supplementary Fig. 2e), we examined the role of p38 in the differentiation of FasL-$T_H$9. Strikingly, Fas ligation induced a much higher frequency of IL-9-producing T cells in cells treated with the p38 inhibitor SB203580 than in untreated cells (Fig. 4a). However, SB203580 alone could not increase c$T_H$9 (Fig. 4a). Additionally, knocking down p38α expression promoted the differentiation of FasL-$T_H$9 but not c$T_H$9 (Supplementary Fig. 4a, b). PKCβ can mediate p38 activation[49]. To elucidate whether PKCβ is involved in Fas ligation-induced p38 activation, we treated WT CD4$^+$ T cells with enzastaurin and found no increased activation of p38 due to Fas ligation (Fig. 4b). These results demonstrated that PKCβ-activated p38 provides negative feedback in the differentiation of FasL-$T_H$9.

NFAT1 cooperates with NF-κB to induce IL-9 transcription in CD4$^+$ T cells[50]. p38 can inhibit NFAT signaling by mediating NFAT phosphorylation[51]. We hypothesized that p38 might

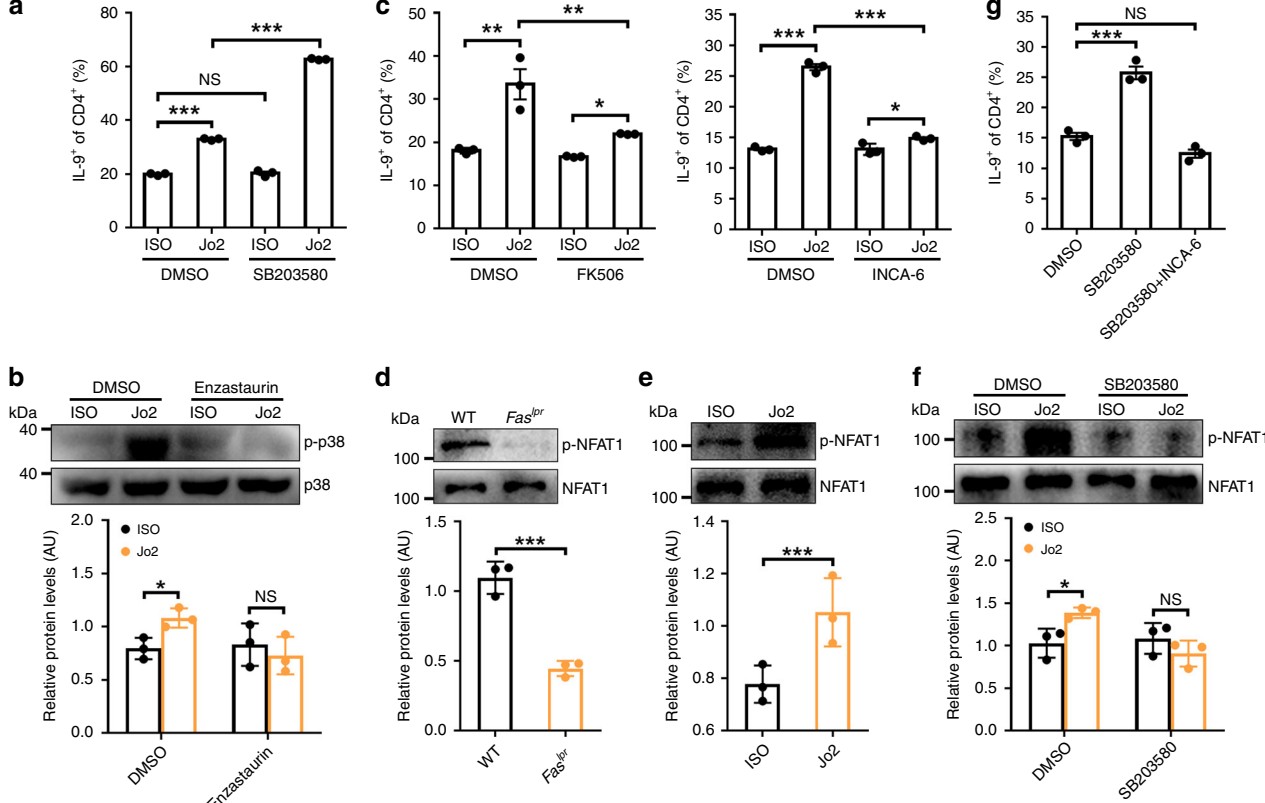

**Fig. 4** p38 inhibits T helper type 9 (T$_H$9) cell generation by Fas signaling. **a** Flow cytometric analysis of IL-9$^+$ (interleukin-9$^+$) cell frequency among CD4$^+$ T cells after naive CD4$^+$ T cells were stimulated with 10 μg ml$^{-1}$ ISO or Jo2 and with or without 0.4 μM p38 inhibitor SB203580 under T$_H$9-skewing conditions for 4 days. **b** Western blotting analysis of p-p38 proteins in naive CD4$^+$ T cells stimulated with 10 μg ml$^{-1}$ ISO or Jo2 for 15 min and with or without 0.5 μM enzastaurin under T$_H$9-skewing conditions (up), and statistical analysis of the relative protein levels (down). **c** Flow cytometric analysis of IL-9$^+$ cell frequency among CD4$^+$ T cells after naive CD4$^+$ T cells were stimulated with 10 μg ml$^{-1}$ ISO or Jo2 and with or without NFAT inhibitor (5 pM FK506 or 50 nM INCA-6) under T$_H$9-skewing conditions for 4 days. **d**, **e** Western blotting analysis of p-NFAT1 proteins in WT or $Fas^{lpr}$ CD4$^+$ T cells (up) (**d**) or WT CD4$^+$ T cells stimulated with 10 μg ml$^{-1}$ ISO or Jo2 (**e**) under T$_H$9-skewing conditions for 15 min (up), and statistical analysis of the relative protein levels (down) (**d**, **e**). **f** Western blotting analysis of p-NFAT1 proteins in WT CD4$^+$ T cells stimulated with 10 μg ml$^{-1}$ ISO or Jo2 and with or without 0.4 μM SB203580 under T$_H$9-skewing conditions for 15 min (up) and statistical analysis of the relative protein levels (down). **g** Flow cytometric analysis of IL-9$^+$ cell frequency among CD4$^+$ T cells after naive CD4$^+$ T cells were stimulated with 10 μg ml$^{-1}$ Jo2 in the presence of 0.4 μM SB203580 or 50 nM INCA-6 plus 0.4 μM SB203580 under T$_H$9-skewing conditions for 4 days. AU, arbitrary units; NS, not significant; *$P < 0.05$, **$P < 0.01$, and ***$P < 0.001$ (unpaired Student's $t$ test (**a**, **c**, **g**)). Representative results from three independent experiments are shown (mean and s.d.). Relative protein levels = the indicated protein gray values/p38 (**b**) or NFAT1 (**d**–**f**) gray values

inhibit the differentiation of FasL-T$_H$9 by regulating NFAT1 phosphorylation. We first determined the role of NFAT in Fas signaling-induced IL-9-producing T cell generation and found that both FK506, an inhibitor of NFAT calcineurin inactivation[52], and INCA-6, a potent and selective inhibitor of calcineurin-NFAT signaling[53], partially inhibited the differentiation of FasL-T$_H$9 (Fig. 4c). Neither inhibitor could restrain the Fas ligation-induced activation of NF-κB (Supplementary Fig. 4c). These results indicated that NFAT1 synergistically enhanced the NF-κB-mediated induction of IL-9-producing T cells by Fas ligation. NFAT1 phosphorylation was markedly reduced in $Fas^{lpr}$ CD4$^+$ T cells under T$_H$9-skewing conditions (Fig. 4d), indicating NFAT1 activation. In contrast, Fas ligation increased NFAT1 phosphorylation in WT CD4$^+$ T cells (Fig. 4e). Then, we examined the effect of p38 on NFAT1 phosphorylation and found that SB203580 treatment abolished the Fas ligation-induced increase in NFAT1 phosphorylation under T$_H$9-skewing conditions (Fig. 4f). Furthermore, SB203580 treatment resulted in no difference in phosphorylated NFAT1 levels between $Fas^{lpr}$ and WT CD4$^+$ T cells under T$_H$9-skewing conditions (Supplementary Fig. 4d). In the presence of INCA-6, SB203580 could no longer

enhance the differentiation of FasL-T$_H$9 (Fig. 4g). These results indicated that p38 limits the differentiation of FasL-T$_H$9 by affecting NFAT1 phosphorylation.

**FasL-T$_H$9 exacerbate murine IBD via IL-9.** IL-9 determines the pathogenesis of ulcerative colitis[10]. We tested the in vivo relevance of our observations in a murine IBD model. The transfer of either cT$_H$9 or FasL-T$_H$9 significantly increased weight loss and shortened colonic length in IBD mice (Fig. 5a, b). However, the ability of the FasL-T$_H$9 to exacerbate IBD was stronger than that of the cT$_H$9 (Fig. 5a, b). Histological analysis also revealed more leukocyte infiltration and more severe damage to glandular structures in the colonic tissue in the mice that received the transfer of FasL-T$_H$9 (Fig. 5c). Moreover, FasL-T$_H$9 caused colonic tissue to produce more IL-6, TNF, IL-1β, IL-10, and IL-22 (Fig. 5d). In contrast, compared with WT-T$_H$9-treated mice, mice that received transfer of $Fas^{lpr}$-T$_H$9 showed reduced weight loss and colonic length shortening (Supplementary Fig. 5a, b). Histological assessment exhibited a similar trend (Supplementary Fig. 5c). These results indicated that Fas ligation exacerbated

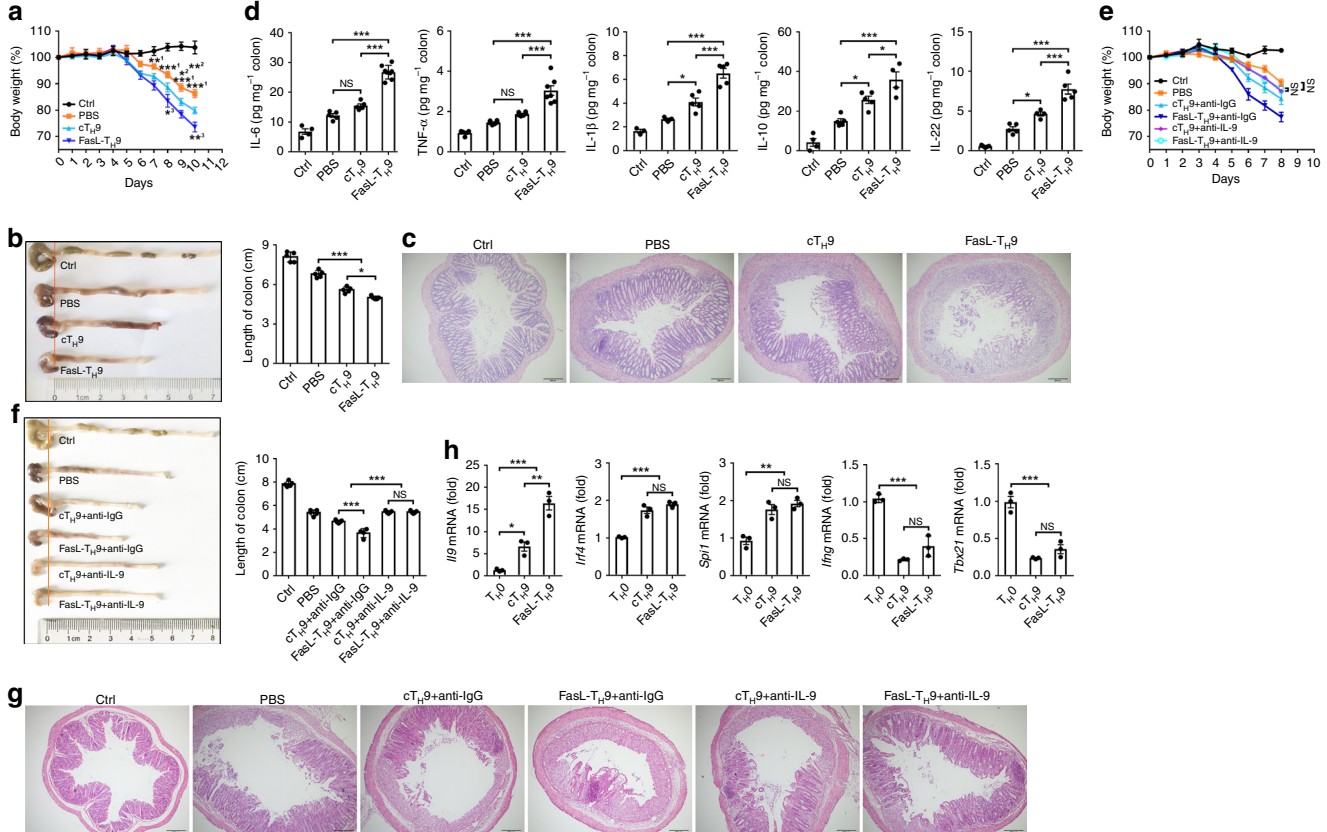

**Fig. 5** FasL-T$_H$9 exacerbate murine inflammatory bowel disease (IBD) via interleukin-9 (IL-9). **a–d** Weight ($n = 5$) (**a**), colonic length ($n = 5$) (**b**), hematoxylin–eosin-stained colonic tissue sections (**c**), and enzyme-linked immunosorbent assay (ELISA) measurement of the indicated cytokines released by colonic tissue ($n = 3$–6) (**d**) of 2.5% dextran sulfate sodium salt (DSS) (w/v)-induced IBD mice that received an intravenous transfer of $2 \times 10^6$ conventionally differentiated T$_H$9 cells (cT$_H$9) or Fas ligand (FasL)-T$_H$9. **e–g** Weight ($n = 5$) (**e**), colonic length ($n = 5$) (**f**), and hematoxylin–eosin-stained of colonic tissue sections (**g**) of 2.5% DSS (w/v)-induced IBD mice that received an intravenous transfer of $2 \times 10^6$ cT$_H$9 or FasL-T$_H$9 plus intravenous injection of 100 μg of anti-IgG- or anti-IL-9-neutralizing antibodies every other day. **h** Real-time PCR measurement of the indicated genes in CD45.1$^+$ cT$_H$9 or FasL-T$_H$9 sorted from lamina propria lymphocytes (LPLs) isolated from 2.5% DSS (w/v)-induced CD45.2$^+$ IBD mice that received an intravenous transfer of $1 \times 10^7$ CD45.1$^+$ cT$_H$9 or FasL-T$_H$9 for 5 days ($n = 3$). Scale bar = 200 μm. NS, not significant; *$P < 0.05$, **$P < 0.01$, and ***$P < 0.001$ (unpaired Student's $t$ test: **a**, **b** (right), **d–f** (right), and **h**). 1, compared with FasL-T$_H$9; 2, compared with cT$_H$9; and 3, compared with cT$_H$9. Representative results from three independent experiments are shown (mean and s.d.)

murine IBD. To test the effects of Fas signaling on T$_H$9 cell apoptosis and proliferation in vivo, we transferred carboxy-fluorescein succinimidyl ester (CFSE)-labeled WT-T$_H$9 or *Fas$^{lpr}$*-T$_H$9 into IBD mice and evaluated apoptosis and proliferation in the CFSE-positive cells in the mesenteric lymph nodes. We found that there were no obvious differences in apoptosis or proliferation between the WT-T$_H$9 and *Fas$^{lpr}$*-T$_H$9 (Supplementary Fig. 5d). These findings excluded the possibility that the apoptosis and proliferation of T$_H$9 cells with Fas defects affect IBD pathogenicity.

To define the effect of IL-9 on T$_H$9 cell-mediated IBD progression, we neutralized IL-9 with anti-IL-9 when cT$_H$9 or FasL-T$_H$9 was transferred. Both types of T$_H$9 cells barely exacerbated murine IBD after IL-9 neutralization (Fig. 5e–g). We also detected similar effects of both types of T$_H$9 cells on IBD *Il9r$^{-/-}$* mice (Supplementary Fig. 5e–g). These results demonstrated that the aggravation of murine IBD by FasL-T$_H$9 depends on IL-9.

Published data suggest that in an inflammatory state in vivo, T$_H$9 cells are unstable and begin to secrete IFN-γ[54]. Therefore, we examined the stability of T$_H$9 cells in IBD mice in vivo by transferring Fas ligation-induced CD45.1$^+$ T$_H$9 cells into CD45.2$^+$ IBD mice. We found that the CD45.1$^+$ cells within the lamina propria lymphocytes (LPLs) maintained their expression of T$_H$9-related genes (*Il9*, *Irf4*, and *Spi1*) and did not exhibit

expression of T$_H$1-related genes (*Ifng* and *Tbx21*) (Fig. 5h). These data indicated that FasL-T$_H$9 are stable in IBD mice in vivo.

**Fas signaling relates to the antitumor activity of T$_H$9 cells.** T$_H$9 cells are present in mice bearing melanoma tumor and exhibit prominent antitumor activity[11,55]. To dissect whether Fas signaling is related to T$_H$9 cells in tumor-bearing mice, we injected B16F10 cells intravenously into WT mice and analyzed *Il9* mRNA levels in Fas-positive and Fas-negative CD4$^+$ T cells. We found that the *Il9* mRNA levels in the Fas$^+$CD4$^+$ T cells were significantly higher than those in the Fas$^-$CD4$^+$ T cells (Supplementary Fig. 6a). In addition, we detected higher *Spi1* and *Irf4* mRNA levels in the Fas$^+$CD4$^+$ T cells than in the Fas$^-$CD4$^+$ T cells and similar *Tbx21*, *Gata3*, *Rorc*, and *Foxp3* mRNA levels between the two cell subsets (Supplementary Fig. 6a), further supporting the idea that Fas signaling dictates T$_H$9 cell differentiation.

To elucidate whether Fas signaling determines the antitumor effects of endogenous T$_H$9 cells, we reconstituted irradiated WT mice with *Fas$^{lpr}$* (*Fas$^{lpr}$* → WT) or WT (WT → WT) bone marrow cells and established tumors with Lewis lung carcinoma cells expressing full-length ovalbumin (LLC-OVA) in these mice. We found that the tumor progression in the *Fas$^{lpr}$* → WT mice

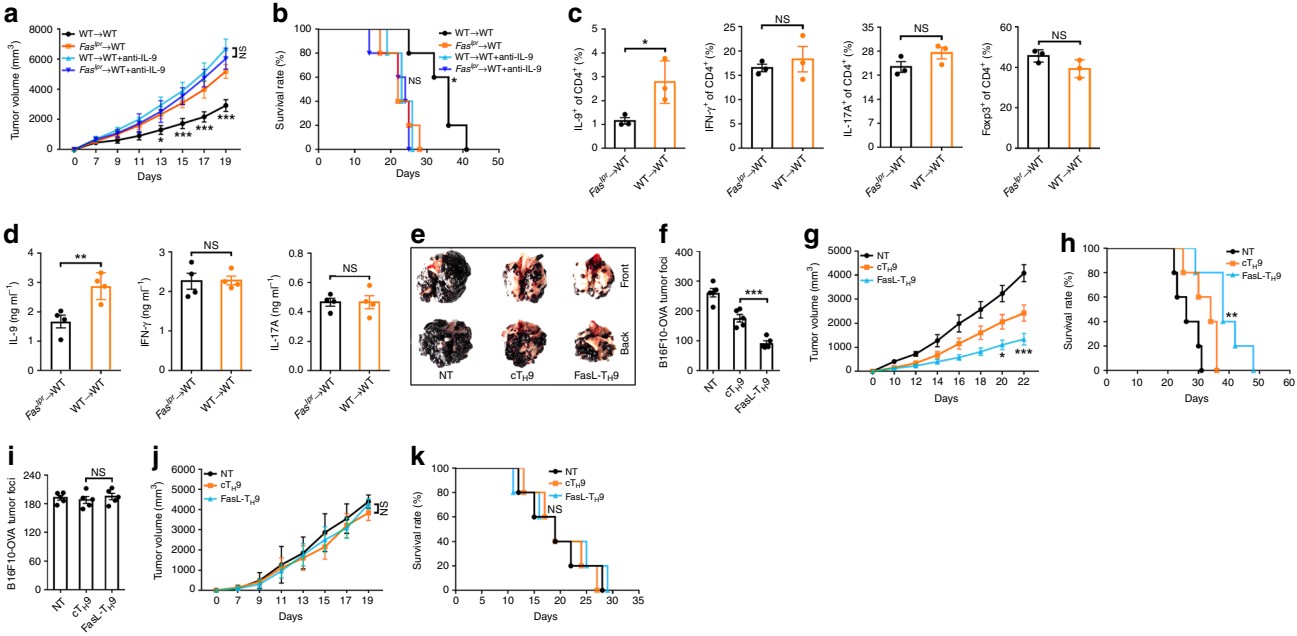

**Fig. 6** Fas signaling relates to the antitumor activity of T helper type 9 (T_H9) cells. (**a**, **b**) Tumor growth (**a**) and survival (**b**) of irradiated WT mice reconstituted with bone marrow cells from wild-type (WT) or *Fas^lpr* mice for 2 months and then subcutaneously injected with LLC-OVA tumor cells with or without intravenous injection of 100 μg of anti-interleukin-9 (IL-9)-neutralizing antibodies every other day (*n* = 5). **c** Flow cytometric analysis of the frequencies of IL-9^+, IFN-γ^+, IL-17A^+, and Foxp3^+ cells among CD4^+ T cells in the tumor-infiltrating lymphocytes (TILs) of the mice described in **a** 20 days after tumor inoculation (*n* = 3). (**d**) Enzyme-linked immunosorbent assay (ELISA) measurements of IL-9, IFN-γ, and IL-17A levels secreted by OVA_{323–339}-stimulated TILs from the mice described in **a** 20 days after tumor inoculation (*n* = 3). **e**, **f** Representative lung appearance (**e**) and statistical analysis of the lung tumor foci (**f**) (*n* = 5) of WT mice 16 days after intravenous injection of B16F10-OVA melanoma cells with no transfer (NT) or transfer of OT-II cT_H9 or FasL-T_H9 1 and 6 days later. **g**, **h** Tumor growth (**g**) and survival (**h**) of WT mice that received a subcutaneous injection of B16F10-OVA cells followed by NT or the intravenous injection of 2 × 10^6 OT-II cT_H9 or FasL-T_H9 1 and 6 days later (*n* = 5). **i** Lung tumor foci of *Il9r^{−/−}* mice 16 days after intravenous injection of B16F10-OVA melanoma cells with NT or transfer of OT-II cT_H9 or FasL-T_H9 1 and 6 days later (*n* = 5). **j**, **k** Tumor growth (**j**) and survival (**k**) of *Il9r^{−/−}* mice that received a subcutaneous injection of B16F10-OVA cells followed by NT or intravenous injection of 2 × 10^6 OT-II cT_H9 or FasL-T_H9 1 and 6 days later (*n* = 5). NS, not significant; *$P < 0.05$, **$P < 0.01$, and ***$P < 0.001$ (unpaired Student's *t* test: **a**, **c**, **d**, **f**, **g**, **i**, and **j**; log-rank test: **b**, **h**, **k**). Compared with the *Fas^lpr* → WT mice in **a**, **b**; compared with cT_H9 in **g**, **h**, **j**, and **k**. Representative results from three independent experiments are shown (mean and s.d.)

was superior to that in the WT → WT mice, which was IL-9 dependent (Fig. 6a, b). The tumor-infiltrating lymphocytes (TILs) in the WT → WT mice demonstrated an enhanced percentage of IL-9-producing CD4^+ T cells but no increased percentages of IFN-γ- or IL-17A-producing CD4^+ T cells or Tregs (Fig. 6c). Restimulation of TILs with OVA_{323–339} in vitro resulted in an evident increase in IL-9 protein expression, but not in IFN-γ or IL-17A protein expression, in the CD4^+ T cells within the TILs of WT → WT mice (Fig. 6d). IL-9 is reported to exert an antitumor effect by activating CD8^+ T cells[55]. We detected an increased frequency of IFN-γ-producing CD8^+ T cells within the TILs of the WT → WT mice (Supplementary Fig. 6b). Moreover, restimulation of TILs with OVA_{257–264} in vitro led to a notable increase in IFN-γ protein levels in the CD8^+ T cells with the TILs of the WT → WT mice (Supplementary Fig. 6c). These results suggested that Fas defects probably restrain antitumor immunity by suppressing IL-9-producing T cell generation.

To directly demonstrate that Fas signaling influences the antitumor properties of T_H9 cells, we differentiated naive OT-II CD4^+ T cells into T_H9 cells with or without Jo2 stimulation. OT-II CD4^+ T cells transgenically express a TCR recognizing an epitope of OVA_{323–339} in the context of I-Ab[12]. We intravenously transferred differentiated OT-II T_H9 cells with OVA-expressing B16F10 cells (B16F10-OVA) into WT mice on the same day and found that FasL-T_H9 exhibited stronger antitumor effects than cT_H9 (Fig. 6e, f). FasL-T_H9 also had superior antitumor effects on

a B16F10-OVA tumor model established by subcutaneous inoculation (Fig. 6g, h). By using LLC-OVA, we obtained similar results (Supplementary Fig. 6d–f). To assess the role of IL-9 in the antitumor effects of FasL-T_H9, we intravenously or subcutaneously challenged *Il9r^{−/−}* mice with B16F10-OVA tumor cells and found that neither FasL-T_H9 nor cT_H9 enhanced antitumor immunity (Fig. 6i–k). Collectively, these results demonstrated that FasL-T_H9 enhance antitumor immunity via IL-9.

**p38 inhibitor exerts antitumor activity by inducing T_H9.** Given that a p38 inhibitor promoted Fas-induced IL-9-producing T cell generation, we investigated whether this p38 inhibitor could also exert antitumor effects dependent on T_H9 cells. To test this, we first detected LLC-OVA tumor progression in WT mice with or without SB203580 treatment. Notable inhibition of tumor progression by SB203580 was observed, which was IL-9 dependent (Fig. 7a, b). The TILs from WT mice that received SB203580 treatment had a significantly higher frequency of IL-9-producing CD4^+ T cells, but not the IFN-γ- or IL-17A-producing CD4^+ T cells (Fig. 7c). Restimulation of the TILs with OVA_{323–339} in vitro induced increased production of the IL-9 protein, but not the IFN-γ or IL-17A protein, in the CD4^+ T cells within the TILs of the WT mice that received SB203580 (Fig. 7d). Importantly, SB203580 could not inhibit tumor progression in *Fas^lpr* → WT mice, suggesting that SB203580 favored T_H9 cell generation via

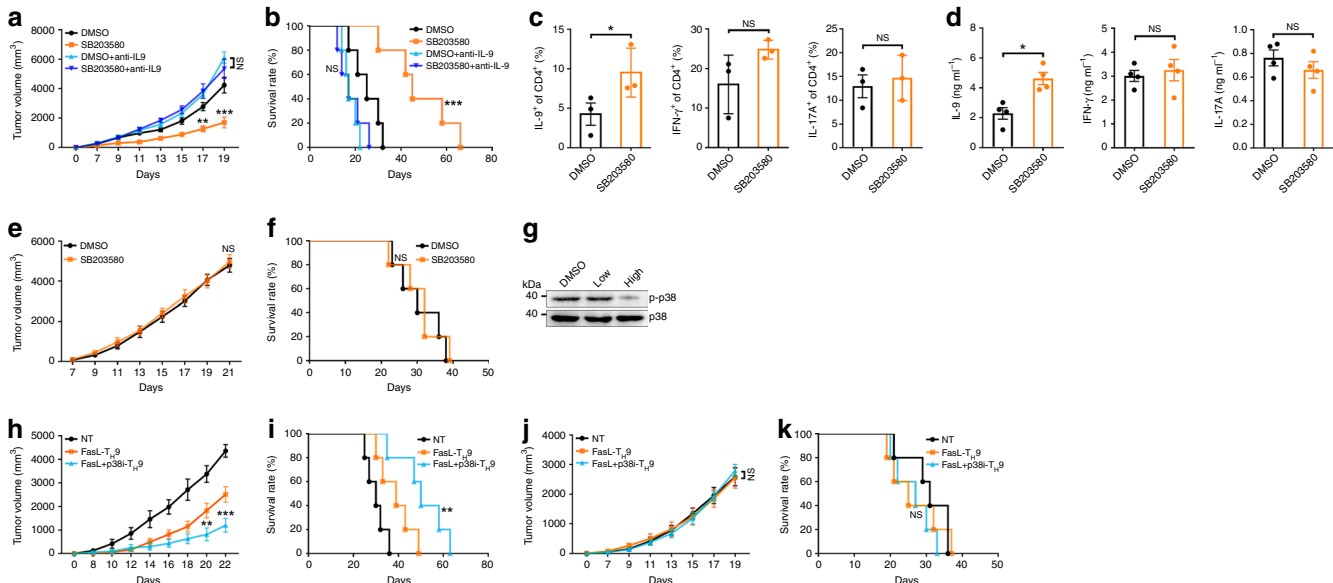

**Fig. 7** p38 inhibitor exerts antitumor activity by inducing T helper type 9 ($T_H9$) cells. **a, b** Tumor growth (**a**) and survival (**b**) of wild-type (WT) mice that received a subcutaneous injection of LLC-OVA cells followed by an intraperitoneal injection of SB203580 (0.5 mg kg$^{-1}$) with or without an intravenous injection of 100 μg of anti-IL-9-neutralizing antibodies every other day ($n = 5$). **c** Flow cytometric analysis of the frequencies of IL-9$^+$ (interleukin-9$^+$), IFN-γ$^+$, and IL-17A$^+$ cells among CD4$^+$ T cells in the TILs of the mice described in **a** 20 days after tumor inoculation ($n = 2$–3). **d** Enzyme-linked immunosorbent assay (ELISA) measurements of IL-9, IFN-γ, and IL-17A levels secreted by OVA$_{323-339}$-stimulated TILs from the mice described in **a** 20 days after tumor inoculation ($n = 3$). **e, f** Tumor growth (**e**) and survival (**f**) of $Fas^{lpr} \rightarrow$ WT mice that received a subcutaneous injection of LLC-OVA cells followed by an intraperitoneal injection of SB203580 (0.5 mg kg$^{-1}$) every other day ($n = 5$). **g** Western blotting analysis of p-p38 expression in colonic tissues from mice that received intraperitoneal injection of low- (0.5 mg kg$^{-1}$) or high-dose (10 mg kg$^{-1}$) SB203580 every other day for a total of 11 injections. **h–k** Tumor growth (**h**) and survival (**i**) of WT mice or tumor growth (**j**) and survival (**k**) of $Il9r^{-/-}$ mice that received a subcutaneous injection of B16F10-OVA cells followed by no transfer (NT) or intravenous injection of 2 × 10$^6$ OT-II FasL-$T_H9$ or FasL + p38i-$T_H9$ 1 and 6 days later ($n = 5$). NS, not significant; *$P < 0.05$, **$P < 0.01$, and ***$P < 0.001$ (unpaired Student's t test: **a**, **c–e**, **h**, **j**; log-rank test: **b**, **f**, **i**, **k**). Compared with dimethyl sulfoxide (DMSO) in **a**, **b**; compared with FasL-$T_H9$ in **h–k**. Representative results from three independent experiments are shown (mean and s.d.)

Fas (Fig. 7e, f). Furthermore, SB203580 did not affect tumor growth in thymus-deficient nu/nu mice (Supplementary Fig. 7a), again highlighting the T cell-dependent antitumor effect. Usually, the dose of SB203580 used for in vivo treatment is 10 mg kg$^{-1}$ (high)[56,57], but the dose we used was 0.5 mg kg$^{-1}$ (low). We dynamically monitored the SB203580 concentration in the plasma of mice that received low- or high-dose SB203580 treatment. The plasma SB203580 concentration reached a peak of 0.15 μg ml$^{-1}$ at 1 h or of 19.46 μg ml$^{-1}$ at 0.5 h in the mice receiving low- or high-dose SB203580 treatment, respectively (Supplementary Fig. 7b). At a dose of 20 μg ml$^{-1}$, but not a dose of 0.15 μg ml$^{-1}$, SB203580 significantly inhibited LLC-OVA cell viability in vitro (Supplementary Fig. 7c). These data further supported the idea of a $T_H9$ cell-dependent antitumor effect of low-dose SB203580.

To evaluate the safety of systemic treatment with low- or high-dose SB203580, we compared the weights, and liver and renal functions of mice that received low-dose SB203580 treatment with those of mice that received high-dose SB203580 treatment. The two groups of mice showed no obvious differences in weight loss or impaired liver and renal functions (Supplementary Fig. 7d, e). Consistently, no pathological injury to the heart or other organs was found in either mouse group (Supplementary Fig. 7f). However, a decrease in the phosphorylated p38 level could be detected only in colonic tissues from the mice receiving high-dose SB203580 treatment (Fig. 7g), suggesting the possibility of systemic inhibition of the p38 pathway. Altogether, our data indicated that low-dose p38 inhibitor treatment restricts tumor progression by enhancing generation of IL-9-producing cells without systemic toxicity in vivo.

Since Fas ligation combined with the p38 inhibitor induced more IL-9-producing T cells, we hypothesized that Fas ligation plus a p38 inhibitor would induce a large number of $T_H9$ cells in the context of adoptive transfer treatment of tumors. To test this, we differentiated naive OT-II CD4$^+$ T cells into $T_H9$ cells with Jo2 stimulation or Jo2 stimulation plus SB203580 treatment (FasL + p38i-$T_H9$). In a B16F10-OVA tumor model established by subcutaneous inoculation, we found that FasL + p38i-$T_H9$ showed stronger antitumor effects than FasL-$T_H9$ (Fig. 7h, i). To elucidate the role of IL-9 in the antitumor effects of these cells, we subcutaneously challenged $Il9r^{-/-}$ mice with B16F10-OVA tumor cells and found that neither FasL + p38i-$T_H9$ nor FasL-$T_H9$ could inhibit tumor progression (Fig. 7j, k). These results indicated that FasL + p38i-$T_H9$ are promising for the adoptive transfer treatment of tumors.

**Fas-related $T_H9$ cells indicate a better prognosis.** To extend our findings to humans, we investigated the effect of Fas ligation on human IL-9-producing T cell induction and found that Fas ligation significantly improved human IL-9-producing T cell generation (Fig. 8a–c). When Fas ligation was combined with a p38 inhibitor, this tendency was more evident (Fig. 8a–c). Then, we detected Fas and IL-9 expression in CD4$^+$ T cells from tumor tissues in a cohort of 36 cancer patients with non-small-cell lung carcinoma (NSCLC) (Supplementary Table 1). Immuno-fluorescence staining revealed that the patients with high numbers of Fas$^+$CD4$^+$ T cells often exhibited high numbers of IL-9$^+$CD4$^+$ T cells (Fig. 8d). Further analysis confirmed the positive correlation between the numbers of Fas$^+$CD4$^+$ T cells and IL-9$^+$CD4$^+$

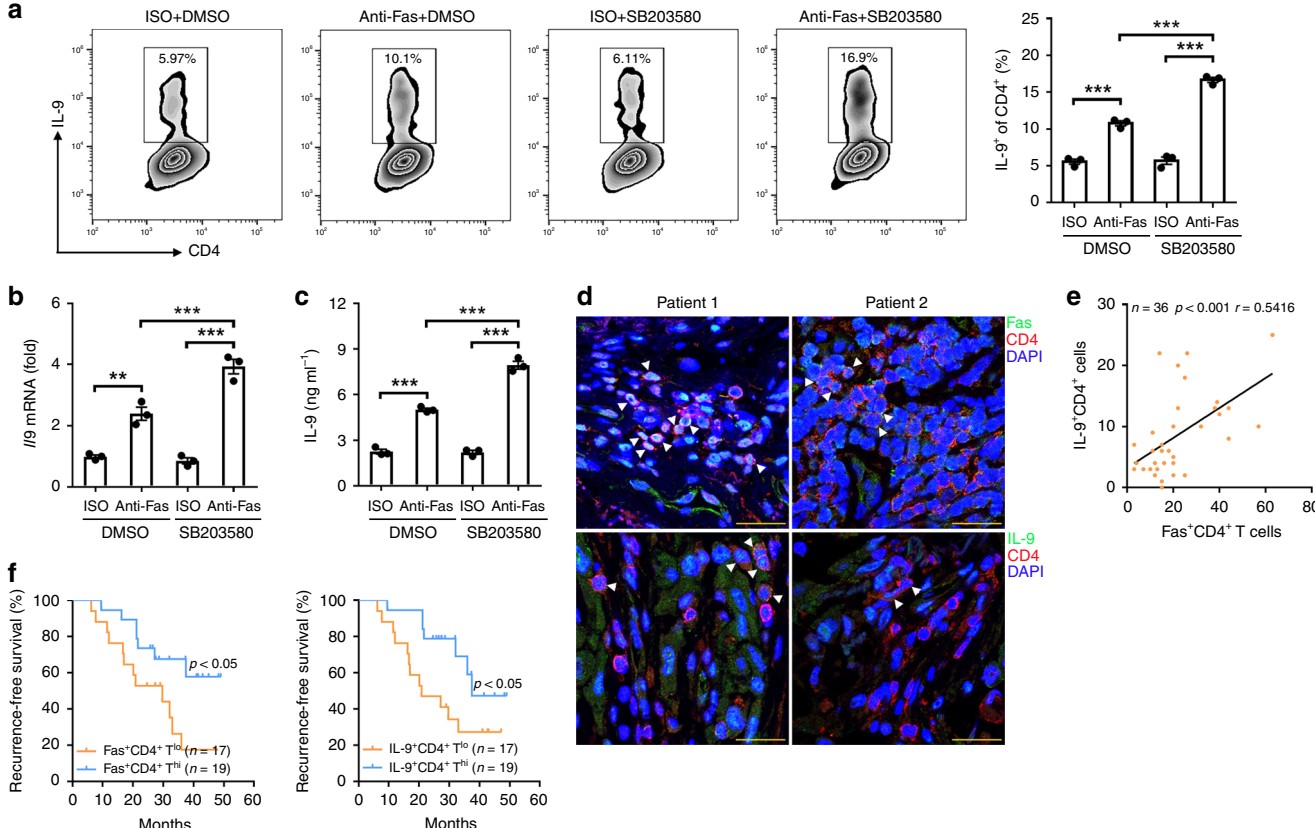

**Fig. 8** Fas-related T helper type 9 (T$_H$9) cells indicate a good prognosis. **a–c** Flow cytometric analysis of the frequency of IL-9$^+$ (interleukin-9$^+$) cells among CD4$^+$ T cells (left) and the corresponding statistical analysis (right) (**a**), real-time PCR analysis of *Il9* messenger RNA (mRNA) expression in T$_H$9 cells (**b**), and enzyme-linked immunosorbent assay (ELISA) measurement of the IL-9 level (**c**) in T$_H$9 cells after the stimulation of human naive CD4$^+$CD45RA$^+$CD45RO$^-$ T cells with 10 μg ml$^{-1}$ ISO or antibodies against human Fas (anti-Fas) with or without the p38 inhibitor SB203580 (0.4 μM) under T$_H$9-skewing conditions for 4 days (*n* = 3). **d** Representative immunofluorescence staining of Fas$^+$CD4$^+$ and IL-9$^+$CD4$^+$ T cells in tumor tissues from non-small-cell lung carcinoma (NSCLC) patients. Arrows indicate Fas$^+$CD4$^+$ or IL-9$^+$CD4$^+$ T cells. **e** Pearson's correlation between Fas$^+$CD4$^+$ T cells and IL-9$^+$CD4$^+$ T cells in tumor tissues from NSCLC patients. **f** The relationship between the recurrence-free survival rate of NSCLC patients and the corresponding Fas$^+$CD4$^+$ or IL-9$^+$CD4$^+$ T cell numbers in tumor tissues. Scale bar = 25 μm. **P < 0.01 and ***P < 0.001 (unpaired Student's t test: **a** (right)–**c**, Spearman's rank-order correlation test: **e**; or log-rank test: **f**). Representative results from three independent experiments are shown (mean and s.d.)

T cells in the tumor tissues (Fig. 8e). To confirm that T$_H$9 cells also have antitumor activity in humans, we divided the patients into high Fas$^+$CD4$^+$ T cell number (Fas$^+$CD4$^+$ T$^{hi}$) and low Fas$^+$CD4$^+$ T cell number (Fas$^+$CD4$^+$ T$^{lo}$) groups or high IL-9$^+$CD4$^+$ T cell number (IL-9$^+$CD4$^+$ T$^{hi}$) and low IL-9$^+$CD4$^+$ T cell number (IL-9$^+$CD4$^+$ T$^{lo}$) groups and found that the recurrence-free survival in both the Fas$^+$CD4$^+$ T$^{hi}$ and IL-9$^+$CD4$^+$ T$^{hi}$ patient groups was better than that in the Fas$^+$CD4$^+$ T$^{lo}$ and IL-9$^+$CD4$^+$ T$^{lo}$ patient groups, respectively (Fig. 8f). Collectively, these results demonstrated that Fas signaling also promotes human IL-9-producing T cell induction, which has a beneficial effect on the outcomes of patients with tumor.

## Discussion
Fas signaling is important in the induction of apoptosis in activated T cells. However, there were no obvious differences in apoptosis or proliferation between WT-T$_H$9 and *Fas$^{lpr}$*-T$_H$9 after transfer into IBD mice. This lack of difference might have been caused by the short course of acute IBD, or the Fas levels in the WT-T$_H$9 might not have been sufficient to trigger apoptotic signaling when we evaluated them. However, Fas-mediated non-apoptotic signaling was observable because the mice receiving the transfer of WT CD4$^+$ T cells differentiated under T$_H$9-skewing conditions showed IBD symptoms with increased severity

dependent on IL-9. Therefore, non-apoptotic Fas signaling is critical to generate IL-9-producing T cells and induce IL-9-mediated physiological functions.

Under T$_H$9-skewing conditions with Fas ligation, TCR signaling activates PKCθ, mediating NF-κB activation. We demonstrated that Fas signaling activated PKCβ, which was necessary for increased NF-κB-dependent IL-9-producing cell induction. PKCθ was not implicated in Fas ligation-mediated IL-9-producing T cell induction because enzastaurin completely abolished this induction. Additionally, activated PKCβ could activate p38, which subsequently phosphorylated NFAT1, leading to NFAT1 inactivation. Since NFAT1 cooperates with NF-κB to promote T$_H$9 cell differentiation, Fas ligation initiates both positive and negative signaling for T$_H$9 cell differentiation. However, Fas signaling still markedly promoted T$_H$9 cell differentiation, indicating the decisive role of NF-κB.

p38 inhibitors have been demonstrated to exert antitumor effects[58,59], but serious side effects resulting from the systemic administration of SB203580 restrain its application. Here, we showed that systemic administration of low-dose SB203580 did not cause obvious side effects. Interestingly, we elucidated that even at very low doses, SB203580 could effectively initiate a T$_H$9 cell-dependent antitumor effect. Moreover, T$_H$9 cells induced by Fas plus SB203580 showed relatively strong antitumor efficacy.

Given the important role of $T_H9$ cells in antitumor immunity, SB203580 treatment is potential strategy for cancer therapy.

After their localizing to the plasma membrane, classic PKCs can be activated by calcium[40]. Both Fas and TCR signaling can initiate $Ca^{2+}$ flux. It is difficult to discriminate between $Ca^{2+}$ flux induced by Fas and that induced by TCR signaling. Fas signaling participated in PLCγ activation and the subsequent increase in $Ca^{2+}$ flux. Fas also interacted with Zap-70, mediating PLCγ activation. The Fas mutations Y224A and Y274A abolished the interaction between Fas and Zap-70 and $T_H9$ cell differentiation induced by Fas ligation. Therefore, $Ca^{2+}$ flux induced by Fas signaling is indispensable for PKCβ activation and subsequent $T_H9$ cell differentiation. The role of the $Ca^{2+}$ flux induced by TCR signaling in the activation of PKCβ needs further study. Interestingly, Tyr284 is located in the YXXL motif of Fas, which is necessary for the binding between Fas and Zap-70[47]. However, according to our results, the Y284A mutation in Fas had no effect on the interaction between Fas and Zap-70 or on the Fas ligation-induced induction of IL-9-producing T cells. This finding shows that Tyr224 and Tyr274, but not Tyr284, are responsible for Fas ligation-induced IL-9-producing T cell generation. Thus, we revealed novel functional Tyr sites in Fas.

Fas has been reported to promote $T_H17$ cell development, but inhibit $T_H1$ cell development by sequestering STAT1[24]. We also detected decreased $T_H17$ cell differentiation and increased $T_H1$ cell differentiation in CD4+ T cells with Fas deficiency in vitro. However, in cells with a Fas defect, we detected reduced $T_H9$ cell generation, but the generation of $T_H17$ and $T_H1$ cells remained unchanged in tumor-bearing mice. This result may be attributable to the different mice or different disease models we used. The same TF can play roles in the differentiation of different $T_H$ cell subsets. STAT1 is also involved in reinforcing $T_H9$ cell development[12]. However, Fas signaling-mediated $T_H9$ cell differentiation is STAT1 independent. Therefore, the different environments that CD4+ T cells encounter determine their differentiation fates. $T_H9$-skewing conditions are probably dominant in tumor-bearing mice.

Previous publications demonstrated IL-9- and IL-21-dependent antitumor effects of $T_H9$ cells[12,55]. In these papers, the authors also verified that $T_H9$ cells inhibit tumor growth by activating CD8+ T cells[12,55]. In another publication, the authors demonstrated that $T_H9$ cells are directly cytotoxic to tumor cells and that the antitumor effects of $T_H9$ cells are mast cell dependent[11]. A recent report showed that IL-9 and CD8+ T cells only slightly affect the antitumor efficacy of $T_H9$ cells, while the *Eomes*-dependent granzyme-mediated cytolytic activity and TRAF6-driven hyperproliferation of $T_H9$ cells are responsible for their antitumor efficacy[13]. Our data suggested that Fas signaling promoted IFN-γ production in CD8+ T cells. Fas signaling also upregulated granzyme expression, which appeared to be *Eomes* independent because a Fas defect induced rather than inhibited *Emos* expression in $T_H9$ cells. Overall, the Fas signaling-mediated antitumor efficacy of $T_H9$ cells is IL-9 dependent. Although the antitumor effect of $T_H9$ cells has been well established in mice[11–13], the antitumor effect of $T_H9$ cells on humans is not well defined. Our results demonstrated that Fas ligation promoted human $T_H9$ cell differentiation. The number of Fas+CD4+ T cells and IL-9+CD4+ T cells in tumor tissues was positively correlated, and higher $T_H9$ cell numbers in patients with tumor indicated a better prognosis than lower $T_H9$ cell numbers. Therefore, our findings provide evidence that $T_H9$ cells also have antitumor effects on humans.

In summary, we demonstrate that Fas signaling promotes $T_H9$ cell differentiation through PKC-β-mediated activation of the NF-κB pathway. At the same time, PKC-β-activated p38 inactivates NFAT1 and abolishes the cooperative effect of NFAT1 on NF-κB, providing negative feedback to Fas-induced $T_H9$ cell differentiation (Supplementary Fig. 8).

## Methods

**Human samples**. Human blood from healthy volunteers and human lung cancer tissue samples were obtained from Zhejiang Cancer Hospital. The collection of human samples was approved by the local Ethical Committee and the Review Board of Zhejiang Cancer Hospital. All the patients were informed of the usage of their tissue samples, and signed consent forms were obtained.

**Mice and cell lines**. Female C57BL/6J (6- to 8-week-old) mice and BALB/C-nu/nu mice were purchased from Joint Ventures Sipper BK Experimental Animal Co (Shanghai, China). $Fas^{lpr}$ and $Fasl^{gld}$ mice were purchased from the Jackson Laboratory (Farmington, CT, USA). $Il9r^{-/-}$[60] mice were kindly provided by Dr. Jean-Christophe Renauld (Université Catholique de Louvain, France). Mice were housed in a specific pathogen-free facility, and experimental protocols were approved by the Animal Care and Use Committee of the School of Medicine, Zhejiang University. Murine B16F10 tumor cells and HEK293 cells were obtained from the American Type Culture Collection (Manassas, VA, USA). B16F10-OVA and LLC-OVA cells were provided by Dr. Qibin Leng (University of Chinese Academy of Sciences, Shanghai, China) and Wei Yang (Southern Medical University, Guangzhou, Guangdong, China), respectively. B16F10 and B16F10-OVA cells were cultured in Dulbecco's modified Eagle's medium (DMEM) supplemented with 10% (v/v) fetal calf serum (Lonza, Basel, Switzerland). LLC-OVA and HEK293 cells were cultured in RPMI-1640 medium supplemented with 10% fetal calf serum. All cells were routinely tested for mycoplasma contamination using the Mycoplasma Detection Kit (Lonza) and were found to be negative.

**In vitro differentiation of T cells**. Naive CD4+CD62L$^{hi}$CD44$^{lo}$ T cells were obtained from the spleen and lymph nodes of mice. Sorted naive CD4+ T cells were routinely 98% pure. The sorted naive CD4+ T cells were stimulated with plate-bound anti-CD3 (145-2C11, 2 µg ml$^{-1}$, Bio X cell, West Lebanon, NH, USA) and anti-CD28 (PV-1, 2 µg ml$^{-1}$, Bio X cell) antibodies, and polarized into effector CD4+ T lymphocyte subsets without cytokines, and with anti-IFN-γ (BE0054, 10 µg ml$^{-1}$, Bio X cell) and anti-IL-4 (BE0045, 10 µg ml$^{-1}$, Bio X cell) ($T_H0$ cells); with IL-12 (130-096-707, 20 ng ml$^{-1}$, Miltenyi Biotec, Bergisch Gladbach, Germany) and anti-IL-4 (10 µg ml$^{-1}$) for $T_H1$ cells; with IL-4 (130-094-061, 20 ng ml$^{-1}$, Miltenyi Biotec) and anti-IFN-γ (10 µg ml$^{-1}$) for $T_H2$ cells; with TGF-β1 (130-095-067, 2 ng ml$^{-1}$, Miltenyi Biotec), IL-4 (20 ng ml$^{-1}$), and anti-IFN-γ (10 µg ml$^{-1}$) for $T_H9$ cells; with TGF-β1 (2 ng ml$^{-1}$), IL-6 (130-094-065, 25 ng ml$^{-1}$, Miltenyi), anti-IFN-γ (10 µg ml$^{-1}$), and anti-IL-4 (10 µg ml$^{-1}$) for $T_H17$ cells; or with TGF-β1 (2 ng ml$^{-1}$), anti-IFN-γ (10 µg ml$^{-1}$), and anti-IL-4 (10 µg ml$^{-1}$) for $T_{reg}$ cells. In some experiments, antibodies against murine Fas (Jo2, 10 µg ml$^{-1}$, Thermo Fisher Scientific, Waltham, MA, USA), z-VAD-fmk (S8102, 1 µM, Selleck, Houston, TX, USA), BAY 11-7082 (S2913, 0.4 µM, Selleck), LY2409881 (S7697, 0.5 µM, Selleck), enzastaurin (S1055, 0.5 µM, Selleck), Go 6983 (S2911, 0.01 µM, Selleck), 2-APB (ab120124, 10 µM, Abcam, Cambridge, MA, USA), XC (ab120914, 10 µM, Abcam), U73122 (S8011, 0.1 µM, Selleck), manoalide (ab141554, 10 µM, Abcam), SB203580 (S1076, 0.4 µM, Selleck), FK506 (S5003, 5 pM, Selleck), or INCA-6 (sc-203160, 50 nM, Santa Cruz, Santa Cruz, CA, USA) was added at the beginning of culture. Cells were classically harvested on day 3 for the detection of cytokines by enzyme-linked immunosorbent assay (ELISA) and real-time PCR analyses. For human in vitro CD4+ T cell differentiation, naive CD4+CD45RA+CD45RO− T cells were isolated from the peripheral blood mononuclear cells of healthy donors with the Human Naive CD4+ T Cell Isolation Kit II (STEMCELL, Vancouver, BC, V6A 1B6, Canada), stimulated with plate-bound anti-CD3 (5 µg ml$^{-1}$, Bio X cell) and anti-CD28 (5 µg ml$^{-1}$, BioLegend, San Diego, CA, USA) antibodies, and polarized into $T_H9$ cells with human TGF-β1 (10 ng ml$^{-1}$) and IL-4 (5 ng ml$^{-1}$) (R&D Systems, Minneapolis, MN, USA).

**Immunoblotting analysis**. Purified naive CD4+ T cells were differentiated into $T_H9$ cells for different times. Then, the cells were pelleted by centrifugation for 5 min at 2000 × g and lysed for 30 min at 4 °C in RIPA buffer (50 mM Tris-HCl (pH 7.5), 150 mM NaCl, 1% NP-40, 0.5% sodium deoxycholate, 0.1% sodium dodecyl sulfate, 1 mM EDTA, and protease inhibitors). Subsequently, the cell lysates were separated by sodium dodecyl sulfate-polyacrylamide gel electrophoresis (10–12%) and transferred onto a polyvinylidene difluoride membrane (Millipore, Billerica, MA, USA). The membrane was blocked with 5% bovine serum albumin (BSA) in TBST buffer and then incubated with primary antibodies overnight at 4 °C. After incubating with the corresponding horse radish peroxidase-conjugated secondary antibodies for 1 h, the chemical signal were developed by NcmECL Ultra (P10100A, P10100B, New Cell & Molecular Biotech Co. Ltd, Suzhou, Jiangsu, China), and then the membrane was scanned using the Tanon 4500 Gel Imaging System. Uncropped blotting scans were presented in the affiliated Source Data file. The antibodies were diluted with NCM universal antibody diluent (WB500D, New Cell & Molecular Biotech Co. Ltd.). The antibodies used and the corresponding dilutions are listed in Supplementary Table 2.

**Ca$^{2+}$ flux.** Sorted naive CD4$^+$ T cells were labeled with 4 µg ml$^{-1}$ Fluo4 (Invitrogen, New York, NY, USA) for 1 h at 37 °C, washed with ice-cold phosphate-buffered saline (PBS), and resuspended in PBS. The labeled cells were stimulated with ISO, Jo2, U73211, or Manoalide and immediately subjected to flow cytometry analysis. Mean fluorescence ratios were plotted after analysis with FlowJo software (TreeStar, Ashland, OR, USA).

**Transfection of siRNA.** Transient small interfering RNA (siRNA) transfection into naive CD4$^+$ T cells was performed in vitro using TransIT-TKO (Mirus Bio, Madison, WI, USA) according to the manufacturer's instructions. Twenty-four hours after transfection, the CD4$^+$ T cells were stimulated with plate-bound anti-CD3 and anti-CD28 antibodies, differentiated into T$_H$9 cells as described above, and cultured for an additional 72 h after analysis with an siRNA specific for murine *Fas*, *Zap70* (sc-29312 or sc-36867, respectively, Santa Cruz), or *p38α* (#6417, Cell Signaling), or NC siRNA (sc-37007, Santa Cruz).

**Real-time PCR.** Total RNA was extracted from T cells using TRIzol reagent (Thermo Fisher Scientific) following the manufacturer's instructions. Complementary DNAs (cDNAs) was synthesized using a cDNA Synthesis Kit (TaKaRa, Kusatsu, Shiga, Japan) following the manufacturer's instructions. Real-time PCR was conducted using SYBR Green (TaKaRa). The following PCR conditions were used: 1 cycle at 95 °C for 30 s, 40 cycles of 95 °C for 5 s, and 60 °C for 34 s. Real-time PCR was performed with an Applied Biosystems 7500 real-time PCR system. The primers used are listed in Supplementary Table 3.

**Measurement of cytokine levels.** After 72 h of polarization, cell culture supernatants were assayed by ELISA to measure the levels of mouse IFN-γ, IL-4, IL-9, and IL-17A (BioLegend) according to the manufacturer's instructions. For intracellular staining, cells were cultured for 3 days, restimulated for 1 additional day, and then stimulated for 4 h at 37 °C in RPMI-1640 medium containing phorbol 12-myristate 13-acetate (50 ng ml$^{-1}$, Sigma-Aldrich, St Louis, MO, USA) and ionomycin (1 µg ml$^{-1}$, Sigma-Aldrich). After staining for surface markers, the cells were fixed and permeabilized according to the manufacturer's instructions (Cytofix/Cytoperm Kit, Thermo Fisher Scientific) and then stained for intracellular products. The following monoclonal antibodies were used for the flow cytometric analyses: fixable viability dye eFluor$^{TM}$ 450- or phycoerythrin-conjugated anti-CD4 and allophycocyanin-conjugated anti-IFN-γ, anti-IL-4, anti-IL-9, anti-IL-17A, or anti-Foxp3. All events were acquired on a BeckmanCoulter DxFLEX flow cytometer equipped with CytExpert experiment-based software (BeckmanCoulter, Inc.), and data were analyzed using FlowJo software (TreeStar). Gating strategies were presented in Supplementary Fig. 9.

**Retroviral infection of CD4$^+$ T cells.** Retroviruses were produced by transfecting Plat-E cells with 7.5 µg of pMX-IRES-GFP, pMX-Fas-WT IRES-GFP, pMX-Fas-T189A IRES-GFP, pMX-Fas-T224A IRES-GFP, pMX-Fas-T274A IRES-GFP, or pMX-Fas-T283A IRES-GFP. Cell culture medium was replaced with fresh medium after 10 h, and the retrovirus-containing supernatant was collected after an additional 72 h. To infect T cells, naive CD4$^+$ T cells were first stimulated with anti-CD3 and anti-CD28 antibodies. At 24- and 36-h time points, the activated T cells were infected for 1 h by centrifugation at 1500 × g with 500 µl of viral supernatant in the presence of 10 µg ml$^{-1}$ polybrene and incubated at 37 °C for an additional 1 h before the cells were removed from the viral supernatant and resuspended in the indicated T cell differentiation medium for 4 days.

**Immunofluorescence and confocal microscopy.** ISO- or Jo2-treated CD4$^+$ T cells incubated with or without 2-APB or XC for 15 min were fixed in prewarmed 4% paraformaldehyde for 30 min and permeabilized with 0.1% Triton X-100 for 10 min. After blocking with 5% BSA, the cells were incubated at 4 °C overnight with anti-Fas (4C3, Cell Signaling) and anti-PKCβ1 (A10-F, Abcam) or anti-PKCβ2 (EPR18104, Abcam) antibodies. Primary antibodies were detected using DyLight 488- and DyLight 549-labeled secondary antibodies (Abcam). Nuclei were stained with 4′,6-diamidino-2-phenylindole (Invitrogen). Human paraffin-embedded lung tumor sections were subjected to immunofluorescence staining. The stained tissue sections were viewed under an Olympus FluoView FV1000 confocal microscope and imaged using an Olympus FluoView version 1.4a viewer (Olympus Corp, Tokyo, Japan). Images of the cells and sections were captured, and positive areas were analyzed.

**RNA-seq analysis.** Total RNA was isolated and reverse transcribed into cDNA to generate an indexed Illumina library, followed by sequencing at the Beijing Genomics Institute (Beijing, China) using a BGISEQ-500 platform. High-quality reads were aligned to a mouse reference genome (GRCm38) by Bowtie2. The expression levels of individual genes were normalized to the FKPM (fragments per kilobase million) reads by RNA-seq by an expectation maximization algorithm. Significant differential expression of a gene was defined as a >2-fold expression difference vs. the control with an adjusted *P* value <0.05. A heat map was analyzed by Gene Ontology using Cluster software and visualized with Java Treeview. DEGs were analyzed by Gene Ontology using the AMIGO and DAVID software. The

enrichment degrees of DEGs were analyzed using Kyoto Encyclopedia of Genes and Genomes annotations.

**Cell viability assay.** LLC-OVA cell viability was measured using a CCK-8 assay according to the manufacturer's instructions. LLC-OVA cells were stimulated with dimethyl sulfoxide (DMSO) or 0.15, 10, 20, or 30 µg ml$^{-1}$ SB203580 for 72 h in a 96-well plate, and 10 µl of CCK-8 (MultiSciences, Hangzhou, Zhejiang, China) was added per well and incubated for 2 h at 37 °C. A multiplate reader was used to measure the absorbance at 450 nm. Cell viability was expressed as the absorbance at 450 nm.

**Chromatographic conditions and data acquisition.** Mice received an intraperitoneal injection of DMSO or 0.5 or 10 mg kg$^{-1}$ SB203580. After 0.5, 1, 3, or 6 h, plasma was collected and centrifuged at 12,000 × g for 10 min, and the supernatants were subjected to chromatographic detection. High-performance liquid chromatography analyses were carried out using Agilent 1100 series LC equipment. The elution system was set up as follows: an Inertil ODS-4 column (4.6 × 250 mm$^2$, 5 µm) was eluted using a water and methanol gradient at 40 °C with a constant flow rate of 1.0 ml min$^{-1}$ (0–20 min, 50–100% methanol; 20–25 min, 100% methanol). The fluorescence detector was set at 325 nm (excitation wavelength) and 408 nm (emission wavelength). Under these chromatographic conditions, the retention time of SB203580 was 13.0 min. The acquired data were processed using the ChemStation for LC 3D software. To obtain chromatograms, data were extracted from the software and plotted in the Microsoft Excel 2010 program.

**Transfer of T$_H$9 cells into IBD mice.** WT and *Il9r*$^{-/-}$ mice were randomized to produce groups with similar average body weights. Acute IBD was induced by administering 2.5% (w/v) DSS (MP Biomedicals, Solon, OH, USA) with a molecular weight of 36,000–50,000 in acidified drinking water for 11 days. The day that the mice started to drink the DSS solution was defined as day 0. For in vivo T$_H$9 cell transfer, WT mice were intravenously injected with 2 × 10$^6$ cT$_H$9, FasL-T$_H$9, WT-T$_H$9, or *Fas*$^{lpr}$-T$_H$9 cells on day 0. For in vivo blockade of IL-9 function, WT mice were intraperitoneally injected with 100 µg of anti-IL-9 (MM9C1, Bio X Cell) every other day. Mouse IgG2a (BE0085, Bio X Cell) was used as the isotype control antibodies.

**Colon tissue culture and isolation of LPLs.** Colons were incised longitudinally and washed four times in Hank's buffered salt solution supplemented with penicillin and streptomycin. One-centimeter-long transverse segments were prepared and cultured in serum-free RPMI-1640 medium supplemented with penicillin, streptomycin, L-glutamine, and nonessential amino acids. After 24 h, the supernatants were collected, and the production of IL-6, TNF, IL-1β, IL-10, and IL-22 was measured using ELISAs (BioLegend).

For the isolation of LPLs, small intestines were cut open longitudinally after removing Peyer's patches and washed with DMEM. Then, the open small intestines were cut into pieces approximately out 5 mm in length, and these pieces were incubated in prewarmed DMEM containing 3% fetal calf serum, 0.2% Hank's solution, 5 mM EDTA, and 0.145 mg ml$^{-1}$ dithiothreitol for 10 min with constant agitation. Then, the small intestine samples were incubated in a solution of 3% DMEM, 0.2% fetal calf serum, 0.025% Hank's solution, 50 mg ml$^{-1}$ DNase, and 75 mg ml$^{-1}$ collagenase II for 5 min, and then the dissociated cells were collected. Finally, the solution containing the digested tissue was passed through a 100-µm cell strainer, and LPLs were isolated on an 80%/40% Percoll (GE Healthcare, Uppsala, Sweden) gradient. The sorted LPLs were applied for the following experiment after washing with PBS.

**Tumor growth experiments.** Bone marrow cells were isolated from *Fas*$^{lpr}$ or WT mice, and 1 × 10$^7$ cells were injected intravenously into C57BL/6J mice that had received sublethal irradiation (400 rad) 1 day before. The chimeric mice were used for tumor inoculation after 6–8 weeks. A total of 1 × 10$^6$ LLC-OVA cells were injected subcutaneously into the chimeric mice or WT mice. In vivo IL-9 neutralization was achieved by intraperitoneal injection of 100 µg of anti-IL-9-neutralizing antibodies (MM9C1, Bio X Cell) every other day following tumor implantation. To evaluate the safety of systematic treatment with low- or high-dose SB203580, mice received intraperitoneal injection of SB203580 (0.5 or 10 mg kg$^{-1}$) every other day. Body weight was monitored, and the heart, liver, spleen, lungs, and kidneys were collected for histopathological analysis 20 days later. For p38 inhibition in vivo, mice received an intraperitoneal injection of SB203580 (0.5 mg kg$^{-1}$) every other day following tumor implantation. Tumor size was monitored every other day by Vernier calipers. According to the criteria of the Animal Care and Use Committee of the School of Medicine, Zhejiang University, when the tumor size was over 8000 mm$^3$, the tumor-bearing mice were euthanized by an intraperitoneal injection of 50 mg kg$^{-1}$ pentobarbital sodium. TILs were prepared by enzymatic digestion with 1 mg ml$^{-1}$ collagenase (Sigma-Aldrich), 0.5 mg ml$^{-1}$ DNase I, and 25 µg ml$^{-1}$ hyaluronidase (Sigma-Aldrich) at 37 °C for 30 min, followed by Percoll (GE Healthcare) gradient purification. The isolated TILs were restimulated with the OVA$_{323–339}$ or OVA$_{257–264}$ peptides at a final concentration of 10 or 20 µg ml$^{-1}$, respectively, in vitro.

To assess the antitumor effects of FasL-$T_H9$ cells, $5 \times 10^5$ B16-OVA or LLC-OVA cells were injected intravenously into C57BL/6 mice on day 0. Then, on days 1 and 6, the mice received an intravenous injection of $2 \times 10^6$ effector OT-II $T_H9$ cells differentiated with or without Jo2. Alternatively, $1 \times 10^6$ B16-OVA cells were injected subcutaneously into C57BL/6 mice on day 0. Then, on days 1 and 6, the mice received an intravenous injection of $2 \times 10^6$ effector OT-II $T_H9$ cells differentiated with or without Jo2 or Jo2 combined with SB203580. Lung tumor foci were enumerated after 16 days in a blinded manner. To assess the role of IL-9 in the antitumor effects of FasL-$T_H9$, $5 \times 10^5$ or $1 \times 10^6$ B16-OVA cells were injected intravenously or subcutaneously into $Il9r^{-/-}$ mice. Then, on days 1 and 6, the mice received an intravenous injection of $2 \times 10^6$ effector OT-II $T_H9$ cells differentiated with or without Jo2. The tumor-bearing mice with lung metastasis were euthanized by an intraperitoneal injection of 50 mg kg$^{-1}$ pentobarbital sodium before the onset of delayed action.

**Histopathology**. Heart, liver, spleen, lungs, kidneys, and intestine were dissected from individual mice in groups and immediately fixed in 10% paraformaldehyde. The heart, liver, spleen, lung, kidney, and intestine samples were subjected to hematoxylin–eosin staining.

**Statistical analyses**. Data are expressed as the mean ± standard deviation. An unpaired Student's $t$ test was used for comparisons between two groups, the log-rank test was used for survival rate analysis, and the Spearman's rank-order correlation test was used for Pearson's correlation analysis using the GraphPad Prism 7.0 software. A difference was considered statistically significant if the $P$ value was <0.05.

**Reporting summary**. Further information on research design is available in the Nature Research Reporting Summary linked to this article.

## Data availability

The data underlying all findings of this study are available from the corresponding author upon reasonable request and are provided as a separate Source Data file. The high-throughput RNA-sequencing data have been deposited in the NCBI Sequence Read Archive under the BioProject accession number SRP159792.

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

## Acknowledgements

This study was supported by the National Key R&D Program of China (2016YFA0501800), the National Key Basic Research Program of China (2015CB943301), the National Natural Science Foundation of China (31770951 and 31670877), and the Hangzhou Health Science and Technology Plan (OO20191170). We thank the Core Facilities, Zhejiang University School of Medicine for technical support.

## Author contributions

Y.S., Z.B.S., X.L.L., Z.Y.M., C.J.L., B.Z., Y.H.C., M.D., J.F.G., Y.M., G.S.Z., and D.Y., performed various experiments; L.A. discussed the manuscript. X.L. analyzed the RNA-sequencing data; Z.J.C. and J.L.W. designed the project and supervised the study; Z.J.C. and Y.Y.S. wrote the manuscript.

## Additional information

**Competing interests:** The authors declare no competing interests.

