## [Peer Review File · Nature Communications]

Reviewers' comments:

Reviewer #1 (Th9, IRF4) (Remarks to the Author):

Title: "Fas signaling-mediated TH9 cell differentiation favors bowel inflammation and antitumor functions" by Yingying Shen, Zhengbo Song, Xinliang Lu, Bei Zhang, Zhenwei Xue, Lionel Apetoh, Xu Li, Chaojie Lu, Zeyu Ma, Jufeng Guo, Danfeng Guo, Gensheng Zhang, Jianli Wang & Zhijian Cai.

In the manuscript by Yingying Shen et al., the authors present a comprehensive analysis about the role of Fas-mediated signaling in mouse and human TH9 cell development and function. In detail, the authors nicely demonstrate the Fas-mediated effect on IL9 gene regulation is dependent on NF- κ B in a PKC- β - and Ca²⁺-dependent manner. By inhibition of p38 the authors demonstrate that Fas-mediated increase in IL-9 production can be further enhanced in an NFAT1-dependent manner. Furthermore, the authors provide in vivo relevance of their findings in a preclinical colitis as well as in two different preclinical tumor models and translate their results into human NSCLC patients. Using the p38 inhibitor SB203580, the authors provide a possible treatment strategy to enhance Fas-mediated IL-9 production and to enhance anti-tumor activity of TH9 cells in vivo. Here, the authors used state-of-the-art in vitro techniques as well as in vivo models to proof the relevance of their findings.

In summary, this manuscript represents a well-written and extensive analysis on the role of Fas-mediated signaling in TH9 cell development and function. However, few concerns rose during the review:

Major critique

1. In Figure 1 the authors demonstrate that naïve CD4⁺ T cells from Fas-lpr mice show reduced expression of IL-9 when compared to WT CD4⁺ T cells upon stimulation in TH9 cell-skewing cytokines TGF- β and IL-4. At the same time, TH1 differentiation and subsequent IFN- γ production seems to be enhanced upon stimulation of naïve CD4⁺ T cells from Fas-lpr mice. In 1994 it was demonstrated that IL-9 production by CD4⁺ T cells is strongly attenuated by IFN- γ (Schmitt et al J Immunol 1994). Although the authors provide mRNA data that IFN- γ seems not to be enhanced in FasL-TH9 cells (supplementary figure 1), the authors should be encouraged to comparatively analyze IL-9 production of WT and FasL-TH9 cells stimulated in the presence of TH9 cell-skewing cytokines and anti-IFN- γ .
2. In figure 1 the authors additionally show that iTreg cell development is reduced in naïve CD4⁺ T cells from Fas-lpr mice. Since IL-2 was shown to be decisively involved in the induction of FOXP3 as well as in the IL-9 production by CD4⁺ T cells, the authors should analyze the contribution of IL-2 to the Fas-mediated enhancement of IL-9 production. In the same vein, do the authors see reduced Treg levels in the tumor models used?
3. In figure 3 the authors used a plethora of different pharmacological inhibitors to interfere with PKC β , PLC and/or NF- κ B. Since these signal transduction pathways are also partially involved in IL-9 production/TH9 cell differentiation by WT CD4⁺ T cells it is astonishing to see, that these inhibitors have no effect on Fas-independent IL-9 production/TH9 cell differentiation. What happens e.g. to the expression of IRF4 under these conditions?

Minor critique

1. On page 8 the authors describe Fas-mediated effects on a transcriptome-wide level and refer to supplementary table 1. However, this table does not include information on differentially expressed genes as stated by the authors.

Reviewer #2 (Th9, cytokine signaling)(Remarks to the Author):

Shen et al describe a role for Fas signaling in the development of IL-9-secreting cells. They demonstrate that Fas has a role in vitro, elucidate signaling pathways downstream of Fas, and demonstrate Fas-dependent IL-9 impacts intestinal inflammation and tumor immunity in adoptive transfer models.

Overall, the data presented are convincing and the story novel and informative. There are however numerous small issues that need to be dealt with.

1. The authors imply but do not specifically state that there is FasL-Fas autostimulation in the in vitro cultures. This should be demonstrated more clearly. First, the authors should show Fas and FasL expression on all of the Th subsets. Can the autostimulation be blocked with Abs against FasL? Can the Th9 cells be cultured at more dilute concentrations (to distinguish paracrine vs. autocrine stimulation) to see if there is still a Fas-dependent response? Could the FasL-Gld cultures be performed in the presence of an autologous cell transfected with FasL to see if the effect can be provided in trans? This point should be more explicitly demonstrated.
2. All graphs should start at "0" on the y-axis. The use of axes that start at other numbers is scattered throughout the report and is misleading.
3. The Fas mutant transduction shown in Supp. Fig. 3 should be done in Lpr cells. Otherwise there is endogenous WT Fas that confounds the interpretation of experiments.
4. For a number of the signaling studies in Figures 3-4, only inhibitors are used to elucidate pathways. While this can be powerful, inhibitors can also have off-target effects. At the very least the authors should show data that indicate the inhibitors are effective for the intended targets at the concentrations used in the report. Are the inhibitors added to the cultures daily or just at the beginning of culture. This should also be clarified.
5. The Th2 differentiation in Fig. 1 is not robust. This is not critical to the report, but if the authors want to show this data they should either alter conditions or perhaps examine the effects on other Th2 cytokines.
6. Is the label used on many axes "IL-9+CD4+ cells" referring to percent of CD4 that are IL-9+ or percent of total cells that are CD4 and IL-9+. This is not clear and since it is a common designation, it should be clarified in text and in the labels.
7. The authors mention other TNFRSF members that have shown the ability to enhance IL-9 including OX40 and GITR, but they omit mention of TL1A/DR3.
8. A supplementary table of genes mentioned in the text related to Figure 2 is not present in the submission materials.
9. For Fig. 2a, log(2) scale might show differences more clearly than the log(10) scale.
10. Western blot data (and confocal images) in Fig. 3 and 4 should be quantified. For the time course Western blots, even examining one time point would be useful.
11. Why does INCA-6 increase IL-9 in Fig. 4g but has the opposite effect in 4c?
12. Can an active NFAT protein bypass induced p38 activity? This would be important to show that

NFAT is a primary target in the pathway described.

13. In Fig. 7C, it appears there are only 2 IFN γ values. Is that correct?

14. The authors should provide a schematic of the pathways elucidated, perhaps as supplementary. Given that they are identifying many mediators and some have negative effects, it would help the reader interpret the data.

15. The article would benefit from making sure that the language is accurate. There are a number of places where data are over-interpreted or conclusions are exaggerated. These include....

Being careful to distinguish between more IL-9-producing cells per population vs. more IL-9 produced per cell. The authors have data to support the former but not the latter.

Being careful to distinguish Th9 differentiation from IL-9 production. They have evidence for the latter, but not much for the former.

In the beginning of the discussion the authors state that Fas-Th9 are more pathogenic. Again, is it that per cell the cells are more pathogenic, or that in a transferred population, there are more IL-9-secreting cells. This requires some precision in language.

First of all, we would like to express our sincere gratitude to the reviewers for your constructive and positive comments. We have performed additional experiments and added new data in the revision. The major changes in the revision have been pointed out in the following text.

Reviewers' comments:

Question 1. In Figure 1 the authors demonstrate that naïve CD4⁺ T cells from Fas-lpr mice show reduced expression of IL-9 when compared to WT CD4⁺ T cells upon stimulation in TH9 cell-skewing cytokines TGF- β and IL-4. At the same time, TH1 differentiation and subsequent IFN- γ production seems to be enhanced upon stimulation of naïve CD4⁺ T cells from Fas-lpr mice. In 1994 it was demonstrated that IL-9 production by CD4⁺ T cells is strongly attenuated by IFN- γ (Schmitt et al J Immunol 1994). Although the authors provide mRNA data that IFN- γ seems not to be enhanced in FasL-TH9 cells (supplementary figure 1), the authors should be encouraged to comparatively analyze IL-9 production of WT and FasL-TH9 cells stimulated in the presence of TH9 cell-skewing cytokines and anti-IFN- γ .

Response: We appreciate the comments and apologize for our carelessness. During the induction of T_H cell subsets, we had added the anti-IFN- γ /anti-IL-4. For T_H9 cells, 10 μ g/ml anti-IFN- γ had been added at the beginning of induction. We have described the protocols for induction of T_H cell subsets more specifically and highlighted the changes in yellow in “**Methods**” section of the revision (lines 513-521). We have also detected the induction of T_H9 cells without anti-IFN- γ and found Fas defect still could restrict T_H9 cell differentiation though differentiated T_H9 cells from WT and *Fas^{lpr}* CD4⁺ T cells were both less than that with anti-IFN- γ (Additional Fig. 1).

Additional Figure 1: Flow cytometric analysis of IL-9 expression in T_H9 cells after naïve WT and *Fas^{lpr}* CD4⁺ T cells were differentiated under T_H9-skewing conditions with or without anti-IFN-γ for 3 days.

Question 2. In figure 1 the authors additionally show that iTreg cell development is reduced in naïve CD4⁺ T cells from *Fas-lpr* mice. Since IL-2 was shown to be decisively involved in the induction of FOXP3 as well as in the IL-9 production by CD4⁺ T cells, the authors should analyze the contribution of IL-2 to the Fas-mediated enhancement of IL-9 production. In the same vein, do the authors see reduced Treg levels in the tumor models used?

Response: We dissected the role of IL-2 in Fas-mediated T_H9 cell differentiation. Firstly, we found that *Il2* mRNA level had no difference between WT-T_H9 and *Fas^{lpr}*-T_H9. Moreover, neither addition of exogenous IL-2 nor neutralizing of endogenous IL-2 rescued the differentiated inferiority of *Fas^{lpr}*-T_H9. These data provide evidences that IL-2 is regardless of Fas-mediated T_H9 cell differentiation. These data have been shown in revised Supplementary Fig. 2b-d and the corresponding description highlighted in yellow has also been added in the revision (lines 164-170). We have also detected the Treg levels in the tumor models and found no decreased Tregs in *Fas^{lpr}*→WT mice. This result has been added in revised Fig. 6c. We have also added the corresponding description highlighted in yellow in the revision (line 315).

Question 3. In figure 3 the authors used a plethora of different pharmacological inhibitors to interfere with PKCb, PLC and/or NF-kB. Since these signal transduction pathways are also partially involved in IL-9 production/TH9 cell differentiation by WT CD4⁺ T cells it is astonishing to see, that these inhibitors have no effect on Fas-independent IL-9 production/TH9 cell differentiation. What happens e.g. to the expression of IRF4 under these conditions?

Response: This is a key point. For these experiments, we had tried a serial concentration of inhibitors. When the concentration was too high, the differentiation of T_H9 cells would be largely inhibited. On the contrary, when the concentration was too low, Fas-mediated differentiation of T_H9 cells would not be affected. We used the optimal concentration at which the inhibitors would not affect the differentiation of T_H9 cells but show influence on Fas-mediated T_H9 cell differentiation. As shown in Fig. 3d and g, inhibitors at the optimal concentration did not affect basic protein levels of p-p65. We have also detected the expression of IRF4 under these conditions and found that inhibitors at the optimal concentration did not obviously affect IRF4 protein levels (Additional Fig. 2).

Additional Figure 2: Western blotting analysis of IRF4 protein levels in naïve CD4⁺ T cells differentiated under T_H9-skewing conditions with or without Jo2 stimulation and in the presence of the indicated inhibitors for 24 h.

Question 4. On page 8 the authors describe Fas-mediated effects on a transcriptome-wide level and refer to Supplementary Table 1. However, this table does not include information on differentially expressed genes as stated by the authors.

Response: We apologize for our carelessness. We have included this table as

Supplementary table 1 in the supplementary information of the revision.

Reviewers' comments:

1. The authors imply but do not specifically state that there is FasL-Fas autostimulation in the *in vitro* cultures. This should be demonstrated more clearly. First, the authors should show Fas and FasL expression on all of the Th subsets. Can the autostimulation be blocked with Abs against FasL? Can the Th9 cells be cultured at more dilute concentrations (to distinguish paracrine vs. autocrine stimulation) to see if there is still a Fas-dependent response? Could the FasL-Gld cultures be performed in the presence of an autologous cell transfected with FasL to see if the effect can be provided in trans? This point should be more explicitly demonstrated.

Response: We have detected the *Fas* and *Fasl* mRNA levels in T_H cell subsets and found that T_H9 cells expressed lower *Fas* but higher *Fasl* genes than other subsets. The ratio of *Fas* gene levels to *Fasl* gene levels was the highest in T_H9 cells. We also found that anti-FasL greatly inhibited T_H9 cell differentiation. In addition, transfection of FasL expressing but not empty vector (EV) rescued decreased T_H9 cell differentiation of CD4⁺ T cells from *Fasl*^{gld} mice. These findings indicated that autoactivated Fas signaling does reinforce T_H9 cell differentiation *in vitro*. We have added these results in revised Supplementary Fig. 1i-k and the corresponding description highlighted in yellow in the revision (lines 121-133). We have also differentiated T_H9 cells at a serial cell density and found that the potential of T_H9 cell differentiation was reversely correlated to cell density (Additional Fig. 3). According to this result, we could not determine the role of paracrine or autocrine stimulation of Fas signaling in T_H9 cell differentiation, because this result suggested that factors from T_H9 cells such as IL-9, IL-10 or IL-21 may be a negative feedback to T_H9 cell differentiation. We are interested in further exploring this phenomenon.

Additional Figure 3: Flow cytometric analysis of IL-9 expression in T_H9 cells differentiated at the indicated cell density.

Question 2. All graphs should start at “0” on the y-axis. The use of axes that start at other numbers is scattered throughout the report and is misleading.

Response: In the revised figures, all graphs are start at “0” on the y-axis except for the graphs of body weight of IBD mice.

Question 3. The Fas mutant transduction shown in Supp. Fig. 3 should be done in Lpr cells. Otherwise there is endogenous WT Fas that confounds the interpretation of experiments.

Response: We appreciate this suggestion. We have transfected Fas mutant vectors into *Fas^{lpr}* mice and found the similar results. The data have been shown in the revised Supplementary Fig. 3h and the corresponding description has been added in the revision highlighted in yellow (line 229).

Question 4. For a number of the signaling studies in Figures 3-4, only inhibitors are used to elucidate pathways. While this can be powerful, inhibitors can also have off-target effects. At the very least the authors should show data that indicate the inhibitors are effective for the intended targets at the concentrations used in the report. Are the inhibitors added to the cultures daily or just at the beginning of culture. This

should also be clarified.

Response: We have confirmed that the inhibitors are effective for the intended targets at the concentrations we used (Additional Fig. 4). The inhibitors were added at the beginning of culture. We have indicated this in the revision highlighted in yellow (line 529).

Additional Figure 4: Assessment of the effect of inhibitors. (a) Western blotting analysis of the indicated phosphorylated protein in naïve CD4⁺ T cells differentiated under T_H9-skewing conditions with or without Jo2 stimulation and in the presence of the corresponding inhibitors for 15 min. (b) Flow cytometric analysis of Ca²⁺ flux in naïve CD4⁺ T cells differentiated under T_H9-skewing conditions with or without Jo2 stimulation and in the presence of 2-APB or XC over time.

Question 5. The Th2 differentiation in Fig. 1 is not robust. This is not critical to the report, but if the authors want to show this data they should either alter conditions or perhaps examine the effects on other Th2 cytokines.

Response: We have replaced this data by a better one in the revised Fig. 1a.

Question 6. Is the label used on many axes “IL-9+CD4+ cells” referring to percent of CD4 that are IL-9+ or percent of total cells that are CD4 and IL-9+. This is not clear and since it is a common designation, it should be clarified in text and in the labels.

Response: It refers to percent of CD4⁺ T cells that are IL-9⁺. We have substituted “IL-9⁺ cell frequency among CD4⁺ T cells” for previous description in the revised figure legends highlighted in yellow. We have also changed the label in the revised figures.

Question 7. The authors mention other TNFRSF members that have shown the ability to enhance IL-9 including OX40 and GITR, but they omit mention of TL1A/DR3.

Response: We have cited this paper and added the corresponding description highlighted in yellow in the revision (lines 65-66).

Question 8. A supplementary table of genes mentioned in the text related to Figure 2 is not present in the submission materials.

Response: We apologize for our carelessness. We have included this table as Supplementary table 1 in the supplementary information of the revision.

Question 9. For Fig. 2a, log(2) scale might show differences more clearly than the log(10) scale.

Response: We have redone this graph with log (2) scale showing no obvious difference to the one with log (10) scale. We have showed the graph with log (2) scale in the revised Fig. 2.

Question 10. Western blot data (and confocal images) in Fig. 3 and 4 should be quantified. For the time course Western blots, even examining one time point would be useful.

Response: We have quantified western blotting data by calculating the gray values in the revised Fig. 3, 4. We have also quantified the confocal images in the revised Fig. 3. The corresponding description has also been added in the revised figure legends of Supplementary Fig. 3 and 4, and highlighted in yellow.

Question 11. Why does INCA-6 increase IL-9 in Fig. 4g but has the opposite effect in 4c?

Response: We apologize for our carelessness. The column of INCA-6 was wrong labeled. It should be labeled by SB203580. We have corrected this error in the revision.

Question 12. Can an active NFAT protein bypass induced p38 activity? This would be important to show that NFAT is a primary target in the pathway described.

Response: As shown in Fig. 4b, PKC β inhibitor Enzastaurin completely abrogated Fas-induced p38 phosphorylation, suggesting Fas-induced p38 activation is PKC β dependent. In addition, there is no kinase domain within NFAT. Therefore, the possibility that NFAT bypass induced p38 activity is very low.

Question 13. In Fig. 7C, it appears there are only 2 IFN γ values. Is that correct?

Response: In that experiment, we failed to isolate TILs from several mice and only got 2 IFN- γ data. We have repeated this experiment and gotten 3 IFN- γ data showing in the revised Fig. 7c.

Question 14. The authors should provide a schematic of the pathways elucidated, perhaps as supplementary. Given that they are identifying many mediators and some

have negative effects, it would help the reader interpret the data.

Response: We have added a schematic of the pathways as Supplementary Fig. 8 in discussion section in the revision. The corresponding description has also been added and highlighted in yellow (lines 478-482).

Question 15. The article would benefit from making sure that the language is accurate. There are a number of places where data are over-interpreted or conclusions are exaggerated. These include....

1. Being careful to distinguish between more IL-9-producing cells per population vs. more IL-9 produced per cell. The authors have data to support the former but not the latter.

Response: We have replaced the description of “IL-9-producing cells” by “frequency of IL-9-producing cells” throughout the revision.

2. Being careful to distinguish Th9 differentiation from IL-9 production. They have evidence for the latter, but not much for the former.

Response: The master transcription factor of T_H9 cells has not been revealed yet, so it is difficult to definitively identify T_H9 cells. Currently, when naive CD4⁺ T cells are induced by IL-6 and TGF-β1, IL-9-producing T cells are generally considered as T_H9 cells. In this study, we also use this method to induce T_H9 cells. We detected the PU.1 and IRF4 closely relating to T_H9 cells and excluded the master transcription factors of other T_H cell subsets to confirm the induction of T_H9 cells. Therefore, we concluded that Fas signaling can promote T_H9 cell differentiation. However, according to your suggestion, in the results where we only detected IL-9-producing cells by flow cytometry, we have changed the description of “T_H9 cell differentiation” to “IL-9-producing cells”.

3. In the beginning of the discussion the authors state that Fas-Th9 are more pathogenic. Again, is it that per cell the cells are more pathogenic, or that in a transferred population, there are more IL-9-secreting cells. This requires some precision in language.

Response: We have changed the description “WT-T_H9 transfer” to “transfer of WT CD4⁺ T cells differentiated under T_H9-skewing conditions” in the revision marked by highlight in yellow (lines 411-412).

REVIEWERS' COMMENTS:

Reviewer #1 (Remarks to the Author):

In the revised manuscript by Yingying Shen and colleagues, "Fas signaling-mediated TH9 cell differentiation favors bowel inflammation and antitumor functions" the authors have adequately responded to my criticisms raised with inclusion of additional data further underlining the role of Fas-mediated signaling in TH9 cell differentiation and in vivo function. Overall the manuscript is greatly improved and I would like to recommend it for publication in Nature Communications.

Reviewer #2 (Remarks to the Author):

The authors have satisfactorily addressed my previous concerns. The manuscript is greatly improved.